# LLM-VTP: LLM-Reasoned Visual Token Pruning for Efficient Multi-modal Video Understanding

## Abstract

In this paper, we introduce LLM-VTP, a visual token pruning method designed to enhance the efficiency of multi-modal video understanding. Large Language Models (LLMs) have shown promising performance in video tasks due to their extended capabilities in comprehending visual modalities. However, the substantial redundancy in video data presents significant computational challenges for LLMs. To address this, we propose a training-free approach that leverages the inherent reasoning abilities of LLMs to selectively prune visual features based on question tokens, thereby optimizing model efficiency. We validate our method across multiple-choice, open-ended, text-generation, and long-form benchmarks. Our results demonstrate that LLM-VTP can prune 80%-90% of tokens while maintaining competitive performance. This highlights its superior effectiveness and efficiency compared to existing pruning methods. The source code will be released to facilitate future research.

## 1 Introduction

Large Language Models (LLMs) (Achiam et al., 2023; Yang et al., 2024; Touvron et al., 2023) have significantly advanced multi-modal understanding owing to their exceptional reasoning capabilities and proficiency in following instructions. Within the realm of video understanding, recent studies (Li et al., 2023c; Lin et al., 2023; Zhang et al., 2023a; Li et al., 2024; 2023a; Xu et al., 2024a; Wang et al., 2024a) have capitalized on the use of pre-trained LLMs as foundational models to address video question-answering tasks. However, the redundancy inherent in video content can lead to significant computational expenses for LLMs due to the quadratic complexity of attention mechanisms. Consequently, effectively reducing the number of video tokens while preserving model performance emerges as an intriguing area of research.

Previous approaches attempt to address this challenge in various ways. LLaMA-VID (Li et al., 2023c) proposes compressing each frame into two distinct tokens: context and content tokens. However, this method necessitates extensive pretraining and fine-tuning phases, which limits its broader applicability with readily available video LLMs. Alternatively, LLaVA-PruMerge (Shang et al., 2024) leverages the correlation between the `[CLS]` token and patch tokens within CLIP (Radford et al., 2021) to identify important visual tokens. Yet, this approach does not consider the relevance of the selected tokens to the questions being asked, potentially selecting tokens that are unrelated to the task at hand. In a related vein, methods like Look-M (Wan et al., 2024) and Elastic Cache (Liu et al., 2024c) employ Key-Value (KV) cache eviction strategies (Zhang et al., 2024; Liu et al., 2024b) to merge the KV cache for multi-modal inputs. These strategies either prioritize retaining text tokens or treat visual and textual tokens equally without explicitly identifying the informative visual tokens. Moreover, eviction-based methods require encoding all visual tokens during the prefilling stage, which becomes inefficient when handling long visual sequences.

Building on the analysis presented above, we identify three essential criteria that an optimal pruning method for multi-modal video understanding ought to meet: (1) It should ideally be training-free, facilitating smooth integration with readily available models while reducing the need for extensive retraining or fine-tuning. (2) It is crucial to retain visual tokens specifically relevant to the given questions. This ensures the model maintains high performance while efficient and mitigates the risk

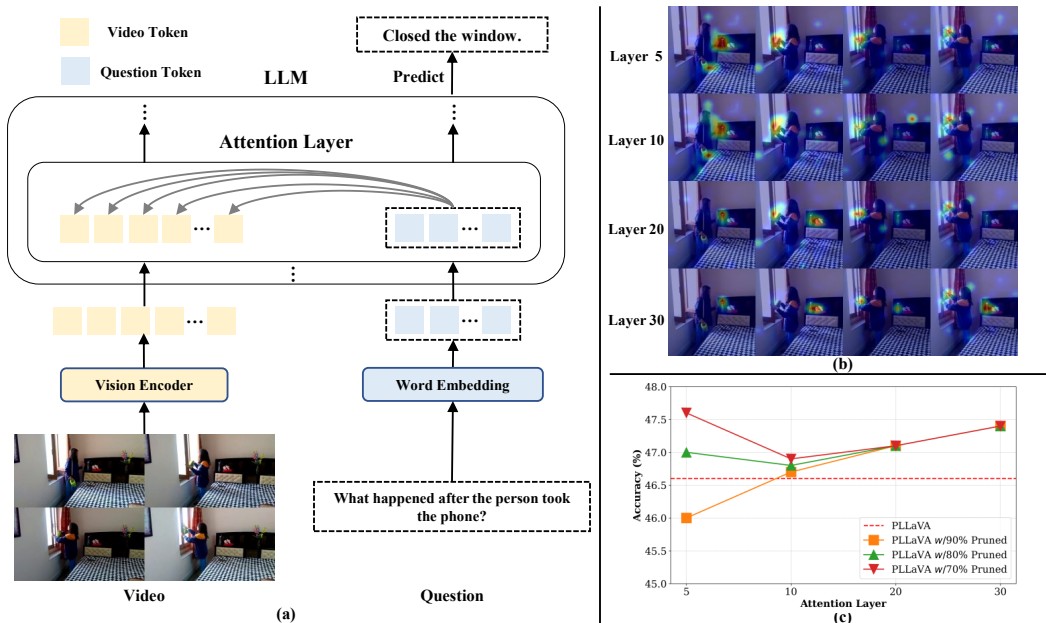

Figure 1: Investigation on studying question-to-video attention characteristics using pre-trained PLLaVA-7B (Xu et al., 2024a). (a) The attention score is calculated by averaging the cross-attention weights between question tokens and each visual token. For simplicity, we omit the system prompt. (b) Visualization of how attention evolves from shallow to deep layers (32 layers in total). The question tokens consistently attend to semantically related visual regions (*e.g.*, the hands and window) throughout different layers. (c) Model performance with token pruning demonstrates that by using the question-to-video attention as guidance, a high ratio of video tokens can be pruned while maintaining competitive performance on MVBench (Li et al., 2024), irrespective of the attention layer's position (*i.e.,* shallow or deep).

of hallucinations when LLMs lack pertinent information (Huang et al., 2024). (3) To save computational costs, the method should avoid the full computation of video tokens during the prefilling stage.

To explore an effective approach that can fulfill these criteria, we investigate the relationship between question tokens and visual tokens using a pre-trained video LLM, *i.e.*, PLLaVA-7B (Xu et al., 2024a). As shown in Fig. 1 (a), we calculate the average attention scores between the question tokens and each visual token, visualizing them from shallow to deep attention layers of the model (Fig. 1 (b)). The visualization reveals a consistent pattern across layers: hand movements and the objects they interact with (e.g., a window) are prominently highlighted, which are highly relevant to the posed question. This phenomenon suggests that critical visual regions can be effectively identified within attention layers, leveraging the powerful reasoning and instruction-following capabilities of LLMs. Based on this, we consider that if the attention weights learned in the shallower layers (e.g., the 5th or 10th layer) could provide clear guidance for distinguishing pivotal tokens from trivial ones, this could serve as an efficient method for video token pruning. To validate this hypothesis, in Fig. 1 (c), we conduct experiments on MVBench (Li et al., 2024) by pruning tokens with the lowest question-to-video attention scores. Our results show that using the attention score from the 5th layer as guidance, pruning up to 90% tokens maintains performance comparable to the baseline model, and reducing the pruning ratio to 80% and 70% leads to even better performance, which demonstrates that LLM is a reliable visual token selector.

Motivated by these insightful observations, we introduce LLM-VTP, a training-free method designed to efficiently prune video tokens by leveraging the discovery ability of LLMs. During the prefilling stage, we utilize the attention weights between question and video tokens within an intermediate layer to identify and retain the discriminative visual tokens that matter for answering the question, while pruning irrelevant ones. Additionally, for the KV caches stored from previous layers, we preserve the selected visual tokens and evict others, thereby reducing computational

costs during the decoding phase. We integrate LLM-VTP with two video LLMs, *i.e.*, PLLaVA (Xu et al., 2024a) and Tarsier (Wang et al., 2024a), and evaluate their performance across the multi-choice Question Answering benchmark (MVBench (Li et al., 2024)), open-ended Video Question Answering benchmarks (MSVD-QA (Chen & Dolan, 2011), MSRVTT-QA (Xu et al., 2016), ActivityNet-QA (Yu et al., 2019), and TGIF-QA (Jang et al., 2017)), text generation benchmark (VideoChatGPT-Bench (Maaz et al., 2023)), and long-from benchmarks (Video-MME (Fu et al., 2024) and Egoschema (Mangalam et al., 2023)). Through comprehensive experimentation, we showcase that LLM-VTP is capable of pruning 80-90% of visual tokens while only experiencing minimal performance degradation in certain instances, and in some cases, it can even bolster model performance. Additionally, our method not only attains competitive outcomes in comparison to the baseline model but also enhances the inference speed, achieving a 1.4 times faster speed on the text generation benchmark.

The main contributions of this paper are as follows:

- We investigate the attention between question and video tokens, discovering that LLMs can effectively locate question-specific visual tokens starting from shallow layers. This insight offers a novel perspective for token pruning in the field of multi-modal video understanding.

- Inspired by our observations, we propose LLM-VTP, a simple yet effective training-free framework. LLM-VTP can be seamlessly integrated with pre-trained video LLMs to efficiently prune video tokens without compromising performance.

- We integrate LLM-VTP with off-the-shelf video LLMs and evaluate its performance across multiple widely used video understanding benchmarks. Our extensive experiments demonstrate that LLM-VTP achieves superior efficiency and effectiveness compared to existing approaches.

## 2 RELATED WORK

### 2.1 VIDEO LARGE LANGUAGE MODEL

Recent advancements in Video LLMs focus on enabling LLMs to comprehend video content. These approaches are broadly categorized into training-free methods and training-required methods.

For training-free approaches (Wu, 2024; Kim et al., 2024; Xu et al., 2024b), they directly adapt the image LLMs for video tasks. FreeVA (Wu, 2024) compacts frame features for LLM processing, and IG-VLM (Kim et al., 2024) merges frames into a single grid, simplifying video-to-image conversion. SF-LLaVA (Xu et al., 2024b) uses a SlowFast (Feichtenhofer et al., 2019) network design, balancing detailed spatial analysis with broad temporal scope efficiently within existing LLM token limits. These methods are ingeniously simple but are limited to handling only brief video clips due to their reliance on the inherent abilities of LLMs to understand temporal sequences.

On the other side, training-required Video LLMs enhance comprehension by training on extensive video datasets. Methods such as Video-ChatGPT (Maaz et al., 2023), Video-LLaVA (Lin et al., 2023), and PLLaVA (Xu et al., 2024a) build upon Image LLMs through video-specific tuning, showing significant improvement in complex video understanding. Other approaches, including VideoChat2 and technologies like VILA (Lin et al., 2024), Tarsier Wang et al. (2024a), Chat-UniVi (Jin et al., 2024), and LLaMA-VID (Li et al., 2023c), introduce efficient token usage, advanced training protocols, or vision-audio understanding, pushing the boundary of what's achievable in video content analysis.

Unlike the methods mentioned above, LLM-VTP aims to enhance the efficiency of existing video LLMs without additional training, which can be applied to training-free or training-required methods.

### 2.2 VISUAL TOKEN PRUNING

Due to the quadratic computational complexity inherent in attention mechanisms, optimizing efficiency through token pruning becomes essential. This optimization highlights a crucial distinction between methods designed for vision-centric and multi-modal tasks.

DynamicViT (Rao et al., 2021) employs a prediction module to selectively prune less important tokens, thereby streamlining the model's efficiency in processing visual data. Similarly, FastViT (Vasu et al., 2023) reduces architectural complexity and memory demands through a novel token mixing operation, catering specifically to vision-only models. Further contributing to this domain, Token Merging (ToMe) (Bolya et al., 2023) merges tokens via token matching, while SPViT (Kong et al., 2022) introduces a method for softly aggregating redundant tokens into a single 'package token', efficiently preserving essential information while minimizing computational load.

Shifting focus from vision-centric methods, LLaVA-Prumerge prunes visual tokens within multi-modal tasks. By applying an adaptive token reduction strategy through PruMerge, which leverages the inherent attention characteristics of CLIP (Radford et al., 2021), LLaVA-Prumerge enhances the efficiency of multi-modal understanding.

However, previous methods primarily focus on directly reducing visual tokens without incorporating textual information for guidance, which may be suboptimal for multi-modal understanding. In contrast, this paper introduces LLM-VTP, which leverages the intrinsic capability of LLMs to identify relevant video tokens for answering questions. This approach effectively enhances model efficiency while maintaining performance.

## 3 METHOD

In this section, we begin by introducing the pre-filling and decoding processes that are fundamental to LLMs handling sequential data. Following this, we describe our proposed method for visual token pruning.

### 3.1 PRELIMINARY

#### 3.1.1 PRE-FILLING STAGE

In the prefilling stage, the model processes the input question tokens and video tokens to construct the initial representations and prepare the KV caches for attention computations. The video tokens are represented as $\boldsymbol{X}_v \in \mathbb{R}^{TN_v \times C}$, where $T$ is the number of frames, and $N_v$ is the number of tokens per frame. Let $\boldsymbol{X}_q \in \mathbb{R}^{N_q \times C}$ denote the question tokens, where $N_q$ is the length of the question and $C$ is the channel dimension. The combined input sequence $X \in \mathbb{R}^{(TN_v + N_q) \times C}$ is formed by concatenating the question and video tokens:

$$X = [\boldsymbol{X}_v; \boldsymbol{X}_q], \tag{1}$$

where $[\cdot; \cdot]$ denotes concatenation along the sequence dimension. The model employs a Transformer architecture with multiple layers. In each layer $l$, the self-attention mechanism computes queries $\boldsymbol{Q}^{(l)}$, keys $\boldsymbol{K}^{(l)}$, and values $\boldsymbol{V}^{(l)}$ through linear projections of the input:

$$\boldsymbol{Q}^{(l)} = X\boldsymbol{W}_Q^{(l)} \quad, \quad \boldsymbol{K}^{(l)} = X\boldsymbol{W}_K^{(l)} \quad, \quad \boldsymbol{V}^{(l)} = X\boldsymbol{W}_V^{(l)}, \tag{2}$$

where $\boldsymbol{W}_Q^{(l)}, \boldsymbol{W}_K^{(l)}, \boldsymbol{W}_V^{(l)} \in \mathbb{R}^{C \times C}$ are learnable weight matrices, and the attention heads are omitted for simplicity. The attention scores are computed using scaled dot-product attention with causal masking to prevent attending to future positions:

$$\boldsymbol{A}^{(l)} = \text{Softmax}\left(\frac{\boldsymbol{Q}^{(l)}(\boldsymbol{K}^{(l)})^\top}{\sqrt{C}} + \boldsymbol{m}\right), \tag{3}$$

where $\boldsymbol{m} \in \mathbb{R}^{(TN_v + N_q) \times (TN_v + N_q)}$ is a causal mask with entries $m_{ij} = -\infty$ if position $i < j$ (future positions) and $0$ otherwise.

The KV caches $\boldsymbol{KV}^{(l)} = (\boldsymbol{K}^{(l)}, \boldsymbol{V}^{(l)})$ are stored for each layer $l$ to facilitate efficient computation during decoding.

### 3.1.2 DECODING STAGE

In the decoding stage, the model generates the answer tokens autoregressively, utilizing the stored KV caches from the pre-filling stage. At each decoding step, given the previously generated tokens,

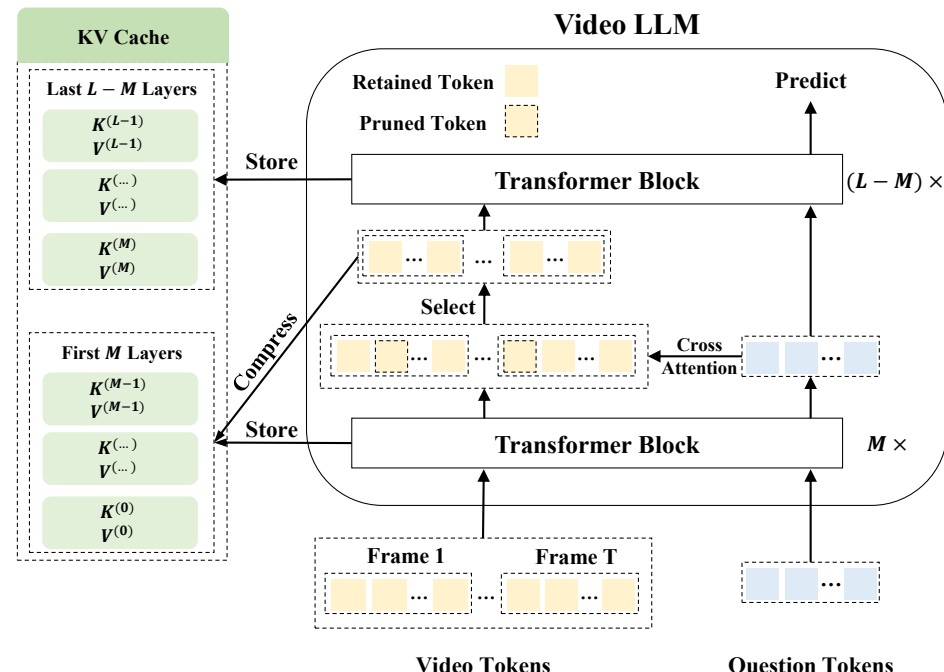

Figure 2: Illustration of the LLM-VTP framework. For simplicity, we omit the system prompt input. By using the question-to-video attention weights learned from an intermediate layer, we can determine which tokens should be pruned to improve efficiency.

the model computes the necessary representations to predict the next token. By using the KV caches, the model efficiently attends to the relevant parts of the input sequence without recomputing the attention for the entire sequence. This process reduces computational overhead and speeds up the generation of the response.

## 3.2 LLM-Reasoned Visual Token Pruning

### 3.2.1 Token Selection

As depicted in Fig. 2, consider a LLM with $L$ layers. During the pre-filling stage, we target the $M$-th layer, where $1 \leq M \leq L$, to compute cross-attention weights between the tokens of the question and the tokens representing the video for obtaining a measure of relevance. Specifically, at the $M$-th layer, we calculate the attention scores $\boldsymbol{A}^{(M)} \in \mathbb{R}^{(TN_v+N_q)\times(TN_v+N_q)}$ according to Equation 3. To obtain the cross-attention scores between question and video tokens, we extract a submatrix $\boldsymbol{A}_{qv}^{(M)} \in \mathbb{R}^{N_q \times TN_v}$ as outlined in the following equation:

$$\boldsymbol{A}_{qv}^{(M)} = \boldsymbol{A}^{(M)}[TN_v :, : TN_v]. \tag{4}$$

Next, we compute the average attention values $\boldsymbol{a}_v \in \mathbb{R}^{TN_v}$ for each video token by averaging over all question tokens and processing these scores on a per-frame basis. For the $t$-th frame, with $1 \leq t \leq T$, the attention scores corresponding to the $N_v$ tokens of that frame are denoted as $\boldsymbol{a}_v^{(t)} \in \mathbb{R}^{N_v}$. We then organize the attention scores for each frame in descending order and select the highest $\alpha\%$ of tokens. The set of indices for the top tokens in frame $t$ is represented by $\mathcal{S}_t$ and is defined as:

$$\mathcal{S}_t = \left\{ j \in \{1, \dots, N_v\} \mid \text{Rank}\left(\boldsymbol{a}_v^{(t)}(j)\right) \leq \lceil \alpha N_v \rceil \right\}, \tag{5}$$

where $\text{Rank}\left(\boldsymbol{a}_v^{(t)}(j)\right)$ indicates the rank of $\boldsymbol{a}_v^{(t)}(j)$ within the sorted attention scores for frame $t$ and $\lceil \rceil$ is the ceiling function. By harvesting the top $\alpha\%$ tokens from each frame, we compile the set

of indices $\mathcal{S}$ for the most significant visual tokens, as given by:

$$\mathcal{S} = \bigcup_{t=1}^{T} \left\{ (t-1)N_v + j \mid j \in \mathcal{S}_t \right\}, \tag{6}$$

which aligns the model's focus with the most question-relevant visual information. To finalize the pre-filling stage, we combine the selected video tokens with the question tokens, thereby enabling processing in the remaining $(L-M)$ layers of the LLM. Subsequently, the Key-Value vectors derived from the retained visual tokens and question tokens, calculated in the last $(L-M)$ layers, are also stored in the KV cache for the decoding process.

### 3.2.2 COMPRESSED KEY-VALUE CACHES

To reduce memory and computational costs during the decoding stage, we compress the KV caches stored from previous $M$ layers by retaining only the selected visual tokens. For each layer $l$ ($1 \leq l \leq M$), the original key and value matrices for the video tokens are $\boldsymbol{K}_v^{(l)} \in \mathbb{R}^{TN_v \times d}$ and $\boldsymbol{V}_v^{(l)} \in \mathbb{R}^{TN_v \times d}$.

We create the compressed key and value matrices $\tilde{\boldsymbol{K}}_v^{(l)}$ and $\tilde{\boldsymbol{V}}_v^{(l)}$ by selecting the rows corresponding to the indices in $\mathcal{S}$:

$$\tilde{\boldsymbol{K}}_v^{(l)} = \boldsymbol{K}_v^{(l)}\left[\mathcal{S},:\right], \quad \tilde{\boldsymbol{V}}_v^{(l)} = \boldsymbol{V}_v^{(l)}\left[\mathcal{S},:\right], \tag{7}$$

where $\boldsymbol{K}_v^{(l)}\left[\mathcal{S},:\right]$ and $\boldsymbol{V}_v^{(l)}\left[\mathcal{S},:\right]$ denote the selection of rows corresponding to the indices in $\mathcal{S}$.

Similarly, we adjust the key and value matrices for the entire sequence by combining the question tokens and the selected visual tokens:

$$\tilde{\boldsymbol{K}}^{(l)} = \left[\tilde{\boldsymbol{K}}_v^{(l)}; \boldsymbol{K}_q^{(l)}\right], \quad \tilde{\boldsymbol{V}}^{(l)} = \left[\tilde{\boldsymbol{V}}_v^{(l)}; \boldsymbol{V}_q^{(l)}\right], \tag{8}$$

where $\boldsymbol{K}_q^{(l)}$ and $\boldsymbol{V}_q^{(l)}$ are the key and value matrices for the question tokens.

By compressing the KV caches, we effectively reduce the sequence length from $N_q + TN_v$ to $N_q + |\mathcal{S}|$, where $|\mathcal{S}|$ represents the total number of selected visual tokens across all frames.

## 4 EXPERIMENT

### 4.1 DATASETS AND EVALUATION METRICS

**Multi-Choice VideoQA.** MVbench (Li et al., 2024) encompasses 20 temporally challenging tasks that cannot be addressed using a single frame. Each task includes 200 test samples, formatted as multiple-choice VideoQA. These samples require the model to choose the correct answer from several provided options.

**Open-Ended VideoQA.** We evaluate our models on four open-ended VideoQA benchmarks: MSVD-QA (Chen & Dolan, 2011), MSRVTT-QA (Xu et al., 2016), ActivityNet-QA (Yu et al., 2019), and TGIF-QA (Jang et al., 2017). To assess the open-ended answers, we utilize GPT-3.5-Turbo-0125 to evaluate both the accuracy (true/false) and quality (scored from 0 to 5) of the responses. The ground truths for these questions are typically single-word answers.

**Text Generation.** VideoChatGPT-Bench, introduced by (Maaz et al., 2023), focuses on five aspects: Correctness of Information (CI), Detail Orientation (DO), Contextual Understanding (CU), Temporal Understanding (TU), and Consistency (CO). The answers for VCGBench are more extensive than those for open-ended benchmarks. For evaluation, we also use GPT-3.5-Turbo-0125 for scoring.

We conduct evaluations of our models using two well-regarded benchmarks for long-form video benchmarks: Video-MME (Fu et al., 2024) and Egoschema (Mangalam et al., 2023). In these evaluations, the models are tasked with selecting the correct answer from multiple-choice options.

### 4.2 BASELINES

To assess the effectiveness of our approach, we employ two state-of-the-art training-free models for comparison, *i.e.*, LLaVA-PruMerge (Shang et al., 2024) and Look-M (Wan et al., 2024). LLaVA-

PruMerge leverages the sparsification of attention scores within CLIP to identify critical tokens and utilizes an outlier detection method to adaptively determine the appropriate pruning ratio. In contrast, Look-M extends the idea of text-only KV cache compression to a multi-modal context by implementing strategies for evicting text-prior KV pairs and merging them with a pivotal merging strategy. For a fair comparison, we utilize the official implementations of these models, adhere to the default settings, and apply them to video benchmarks.

### 4.3 IMPLEMENTATION DETAILS

All experiments are conducted using NVIDIA A100 GPUs with 40GB memory. We implement the proposed LLM-VTP, LLaVA-PruMerge, and Look-M on two video LLMs, specifically PLLaVA (Xu et al., 2024a) and Tasier (Wang et al., 2024a). Regarding the pruning ratio settings, LLaVA-PruMerge can adaptively determine its pruning ratio, so we focus on the settings for LLM-VTP and Look-M. For the multi-choice benchmark, we set the token selection ratio $\alpha$ to 0.1 and the attention calculation $M$ layer to 10 for LLM-VTP based on PLLaVA, and to 0.2 and 10 for LLM-VTP based on Tasier. For Look-M, these configurations are roughly equivalent to setting both the recent and important ratios to 0.05 and 0.1 for the PLLaVA-based and Tarsier-based models, respectively. This is because the number of text tokens is significantly smaller compared to video tokens. For the open-ended and text generation benchmarks, $\alpha$ and $M$ are set to 0.2 and 5 for LLM-VTP based on PLLaVA, and to 0.2 and 15 for LLM-VTP based on Tarsier. Meanwhile, the recent and important ratios are set to 0.1 for Look-M on these benchmarks.

### 4.4 MAIN RESULT

**Multiple-Choice QA.** As shown in Tab. 1, our approach demonstrates better average performance than the original models, even after removing 90% of visual tokens from PLLaVA and 80% from Tarsier. Achieving such performance is quite challenging for other models. Specifically, while PruMerge and Look-M deliver good results on PLLaVA, they struggle to maintain this high accuracy with more powerful models like Tarsier. This highlights the adaptability of our method across different models. Furthermore, it can be observed that PruMerge removes a limited number of tokens and does not achieve better performance compared to our method. This further confirms that our model is more effective at identifying useful tokens.

**Text Generation.** As shown in Tab. 2, on the Video-ChatGPT Bench, our method achieves similar accuracy to PruMerge while using only 20% of the tokens, compared to PruMerge's 34.7%. In contrast, Look-M's accuracy significantly drops when using 20% of the tokens.

**Open-ended QA.** As shown in the Tab. 2, our method, based on two different models, achieves the best performance on almost every metric across four datasets. This further demonstrates the practicality of our approach on various datasets. In contrast, PruMerge and Look-M exhibit a lack of robustness. For example, using the Tarsier model, although Prumerge achieves good accuracy on ActivityNet-QA and TGIF-QA, its performance on MSVD and MSRVTT largely drops. A similar situation applies to Look-M; it performs well on MSVD-QA and TGIF-QA, but its accuracy significantly declines on MSRVTT and ActivityNet-QA.

Additionally, we provide a more comprehensive analysis to demonstrate the superiority of our method in Tab. 9 in Appendix A.1.

### 4.5 DIAGNOSTIC STUDY

**Efficiency Analysis.** In Tab. 3, we present a comparative analysis of the efficiency of various token pruning methods, utilizing the VideoChatGPT-Bench dataset for this evaluation. This dataset is particularly suitable for our study as it tends to yield longer output texts, effectively highlighting the efficiency disparities among the examined methods.

Given that the average output lengths across these methods on the selected dataset are notably similar, we chose to directly compare their average inference times per sample. Our analysis reveals that the Look-M method has the lowest inference efficiency. This decrease in efficiency can be attributed to Look-M's need to compute attention scores across every attention layer, coupled with its layer-wise optimization strategy for caching, which collectively decelerates the process. Conversely,

Table 1: Performance comparison on MVBench, where a retained ratio of 100% indicates that no token pruning method is used. The best results are **bolded** and the second-best results are underlined.

| Method | Retained Ratio | LLM Size | AS | AP | AA | FA | UA | OE | OI | OS | MD | AL | ST | AC | MC | MA | SC | FP | CO | EN | ER | CI | Avg. |
|---|---|---|---|---|---|---|---|---|---|---|---|---|---|---|---|---|---|---|---|---|---|---|---|
| Video-LLaMA (Zhang et al., 2023a) | 100% | 7B | 27.5 | 25.5 | 51.0 | 29.0 | 39.0 | 48.0 | 40.5 | 38.0 | 22.5 | 22.5 | 43.0 | 34.0 | 22.5 | 32.5 | 45.5 | 32.5 | 40.0 | 30.0 | 21.0 | 37.0 | 34.1 |
| LLaMA-Adapter (Zhang et al., 2023b) | 100% | 7B | 23.0 | 28.0 | 51.0 | 30.0 | 33.0 | 53.5 | 32.5 | 33.5 | 25.5 | 21.5 | 30.5 | 29.0 | 22.5 | 41.5 | 39.5 | 25.0 | 31.5 | 22.5 | 28.0 | 32.0 | 31.7 |
| Video-ChatGPT (Maaz et al., 2023) | 100% | 7B | 23.5 | 26.0 | 62.0 | 22.5 | 26.5 | 54.0 | 28.0 | 40.0 | 23.0 | 20.0 | 31.0 | 30.5 | 25.5 | 39.5 | 48.5 | 29.0 | 33.0 | 29.5 | 26.0 | 35.5 | 32.7 |
| VideoChat (Li et al., 2023a) | 100% | 7B | 33.5 | 26.5 | 56.0 | 33.5 | 40.5 | 53.0 | 40.5 | 30.0 | 25.5 | 27.0 | 48.5 | 35.0 | 20.5 | 42.5 | 46.0 | 26.5 | 41.0 | 23.5 | 23.5 | 36.0 | 35.5 |
| VideoChat2 (Li et al., 2024) | 100% | 7B | 66.0 | 47.5 | 83.5 | 49.5 | 60.0 | 58.0 | 71.5 | 42.5 | 23.0 | 23.0 | 88.5 | 39.0 | 42.0 | 58.5 | 44.0 | 49.0 | 36.5 | 35.0 | 40.5 | 65.5 | 51.1 |
| ST-LLM (Liu et al., 2024a) | 100% | 7B | 66.0 | 53.5 | 84.0 | 44.0 | 58.5 | 80.5 | 73.5 | 38.5 | 42.5 | 31.0 | 86.5 | 36.5 | 56.5 | 78.5 | 43.0 | 44.5 | 46.5 | 34.5 | 41.5 | 58.5 | 54.9 |
| PLLaVA (Xu et al., 2024a) | 100% | 7B | 58.0 | 49.0 | 55.5 | 41.0 | 61.0 | 56.0 | 61.0 | 36.0 | 23.5 | 26.0 | 82.0 | 39.5 | 42.0 | 52.0 | 45.0 | 42.0 | 53.5 | 30.5 | 48.0 | 31.0 | 46.6 |
| PLLaVA w/PruMerge+ (Shang et al., 2024) | 36.5% | 7B | 55.0 | 50.5 | 60.0 | 44.5 | 60.5 | 54.5 | 61.5 | 33.0 | 19.5 | 24.5 | 82.0 | 41.5 | 38.0 | 55.0 | 2.5 | 40.5 | 49.5 | 29.5 | 46.0 | 30.5 | 45.9 |
| PLLaVA w/Look-M (Wan et al., 2024) | 10.0% | 7B | 58.5 | 49.5 | 61.0 | 42.5 | 61.0 | 56.5 | 63.5 | 33.5 | 21.0 | 29.0 | 83.5 | 36.0 | 37.5 | 54.5 | 46.5 | 42.5 | 49.5 | 26.5 | 48.5 | 30.0 | 46.5 |
| PLLaVA w/Ours | 10.0% | 7B | 58.5 | 52.0 | 64.0 | 43.0 | 62.5 | 56.5 | 64.5 | 33.5 | 22.5 | 27.0 | 85.0 | 37.0 | 37.0 | 55.0 | 45.0 | 41.0 | 48.0 | 26.5 | 47.5 | 30.5 | 46.9 |
| Tarsier (Wang et al., 2024a) | 100% | 7B | 95.5 | 43.5 | 49.0 | 64.0 | 81.0 | 63.0 | 51.0 | 31.0 | 40.0 | 60.5 | 94.5 | 73.0 | 44.0 | 55.5 | 79.0 | 33.5 | 56.0 | 85.0 | 58.0 | 62.6 |
| Tarsier w/PruMerge+ (Shang et al., 2024) | 36.7% | 7B | 90.0 | 40.5 | 44.0 | 59.0 | 76.0 | 58.0 | 44.0 | 30.5 | 40.0 | 48.0 | 50.5 | 90.0 | 68.0 | 40.0 | 90.5 | 74.5 | 30.0 | 54.0 | 84.0 | 54.5 | 58.3 |
| Tarsier w/Look-M (Wan et al., 2024) | 20% | 7B | 93.0 | 43.5 | 50.0 | 64.0 | 79.0 | 66.0 | 48.5 | 31.5 | 40.0 | 52.5 | 62.0 | 91.5 | 68.0 | 44.0 | 94.0 | 76.0 | 30.5 | 54.5 | 83.0 | 58.5 | 61.5 |
| Tarsier w/LLM-VPT (Ours) | 20% | 7B | 93.5 | 44.5 | 52.0 | 65.0 | 80.0 | 65.0 | 48.0 | 31.0 | 43.0 | 51.5 | 64.0 | 93.5 | 71.0 | 48.0 | 95.0 | 79.0 | 32.5 | 56.5 | 84.0 | 58.5 | 62.8 |

Table 2: Performance comparison on video question-answering benchmarks, where a retained ratio of 100% indicates that no token pruning method is used. The retained ratios for PruMerge exhibit only minor variances across different datasets. Consequently, we average the values across these datasets. The best results are **bolded** and the second-best results are underlined.

| Method | Retained Ratio | LLM Size | Video-ChatGPT Bench | | | | | | MSVD-QA | | MSRVTT-QA | | ActivityNet-QA | | TGIF-QA | |
|---|---|---|---|---|---|---|---|---|---|---|---|---|---|---|---|---|
| | | | CI | DO | CU | TU | CO | Avg. | Acc. | Sco. | Acc. | Sco. | Acc. | Sco. | Acc. | Sco. |
| Video-LLaMA (Zhang et al., 2023a) | 100% | 7B | 1.96 | 2.18 | 2.16 | 1.82 | 1.79 | 1.98 | 51.6 | 2.5 | 29.6 | 1.8 | 12.4 | 1.1 | - | - |
| LLaMA-Adapter (Zhang et al., 2023b) | 100% | 7B | 2.03 | 2.32 | 2.30 | 1.98 | 2.15 | 2.16 | 54.9 | 3.1 | 43.8 | 2.7 | 34.2 | 2.7 | - | - |
| Video-ChatGPT (Maaz et al., 2023) | 100% | 7B | 2.50 | 2.57 | 2.69 | 2.16 | 2.20 | 2.42 | 64.9 | 3.3 | 49.3 | 2.8 | 35.2 | 2.7 | 51.4 | 3.0 |
| Video-LLaVA (Lin et al., 2023) | 100% | 7B | - | - | - | - | - | - | 70.7 | 3.9 | 59.2 | 3.5 | 45.3 | 3.3 | 70.0 | 4.0 |
| Chat-UniVi (Jin et al., 2024) | 100% | 7B | 2.89 | 2.91 | 3.46 | 2.89 | 2.81 | 2.99 | 65.0 | 3.6 | 54.6 | 3.1 | 45.8 | 3.2 | 60.3 | 3.4 |
| MovieChat (Song et al., 2024) | 100% | 7B | 2.76 | 2.93 | 3.01 | 2.24 | 2.42 | 2.67 | 75.2 | 3.8 | 52.7 | 2.6 | 45.7 | 3.4 | - | - |
| VideoChat (Li et al., 2023a) | 100% | 7B | 2.23 | 2.50 | 2.53 | 1.94 | 2.24 | 2.29 | 56.3 | 2.8 | 45.0 | 2.5 | 26.5 | 2.2 | 34.4 | 2.3 |
| VideoChat2 (Li et al., 2024) | 100% | 7B | 3.02 | 2.88 | 3.51 | 2.66 | 2.81 | 2.98 | 70.0 | 3.9 | 54.1 | 3.3 | 49.1 | 3.3 | - | - |
| Vista-LLaMA (Ma et al., 2024) | 100% | 7B | 2.44 | 2.64 | 3.18 | 2.26 | 2.31 | 2.57 | 65.3 | 3.6 | 60.5 | 3.3 | 48.3 | 3.3 | - | - |
| ST-LLM (Liu et al., 2024a) | 100% | 7B | 3.23 | 3.05 | 3.74 | 2.93 | 2.81 | 3.15 | 74.6 | 3.9 | 63.2 | 3.4 | 50.9 | 3.3 | - | - |
| PLLaVA (Xu et al., 2024a) | 100% | 7B | **3.21** | **2.86** | **3.62** | 2.33 | **2.93** | **3.12** | 76.6 | 4.1 | 62.0 | 3.5 | **56.3** | 3.5 | **77.5** | 4.1 |
| PLLaVA w/PruMerge (Shang et al., 2024) | 35.8% | 7B | 3.17 | 2.78 | 3.50 | 2.37 | 2.91 | 3.07 | 77.5 | 4.1 | 61.4 | 3.5 | 55.8 | 3.5 | 77.1 | 4.2 |
| PLLaVA w/Look-M (Wan et al., 2024) | 20.0% | 7B | 3.05 | 2.69 | 3.39 | 2.31 | 2.72 | 2.97 | 76.5 | 4.1 | 62.0 | 3.5 | 55.0 | 3.5 | 76.8 | 4.1 |
| PLLaVA w/LLM-VPT (Ours) | 20.0% | 7B | 3.16 | 2.80 | 3.48 | **2.39** | 2.91 | 3.08 | **77.6** | 4.1 | **62.2** | 3.6 | 55.9 | 3.5 | 76.9 | 4.2 |
| Tarsier (Wang et al., 2024a) | 100% | 7B | **3.53** | **3.00** | **3.82** | **2.74** | **3.10** | **3.37** | 77.0 | 4.1 | 62.0 | 3.5 | 59.5 | 3.6 | 79.2 | 4.2 |
| Tarsier w/PruMerge+ Shang et al. (2024) | 34.7% | 7B | 3.48 | 2.94 | 3.76 | 2.73 | 2.88 | 3.31 | 76.9 | 4.1 | 58.5 | 3.4 | 59.3 | 3.6 | 78.1 | 4.2 |
| Tarsier w/Look-M Wan et al. (2024) | 20.0% | 7B | 3.05 | 2.58 | 3.34 | 2.40 | 2.49 | 2.91 | 78.6 | 4.1 | 61.0 | 3.5 | 57.6 | 3.6 | 78.3 | 4.2 |
| Tarsier w/LLM-VPT (Ours) | 20.0% | 7B | 3.48 | 2.91 | 3.74 | 2.66 | 3.07 | 3.31 | **79.2** | 4.1 | **62.2** | 3.5 | 59.3 | 3.6 | 78.7 | 4.2 |

our model demonstrates a speed enhancement of up to 1.4 times while maintaining high accuracy levels. Our approach also demonstrates superior computational efficiency. Although PruMerge+ achieves comparable computational efficiency to ours, it relies on the [CLS] token for token compression, which is incompatible with some video LLMs based on adapted projectors (*e.g.*, ST-LLM and VideoChat2 with Q-former architecture, see Tab. 8 in Appendix A.1), thus limiting its applicability.

**Ablation Study on $\alpha$ and the Position of Attention Weight Utilization.** In Fig. 3 and Fig. 4, we investigate the impact of using different $\alpha$ values and attention weights from various layers $M$ on four different benchmarks (MVBench, VideoMME, MSVD-QA, and VCG-Bench) with two models (PLLaVA and Tarsier). Although selecting different values for M leads to some mild variations, there is a consistent trend across different datasets: setting $M$ to 5 or higher yields results nearly equivalent to those of the baseline model. These results demonstrate that starting from $M = 5$, the

Table 3: Efficiency comparison of different pruning methods on VideoChatGPT Bench. The inference time is calculated by averaging the test time across all samples.

| Method | Average Decoding Length | Inference Time (s) | Speed Up | FLOPs (T) |
|---|---|---|---|---|
| PLLaVA | 72.5 | 3.12 | - | 26.8 |
| PLLaVA w/PruMerge+ | 72.0 | 2.67 | 1.2× | 9.2 |
| PLLaVA w/Look-M | 72.0 | 6.04 | 0.5× | 26.7 |
| PLLaVA w/LLM-VPT (Ours) | **73.0** | **2.24** | **1.4×** | **8.8** |

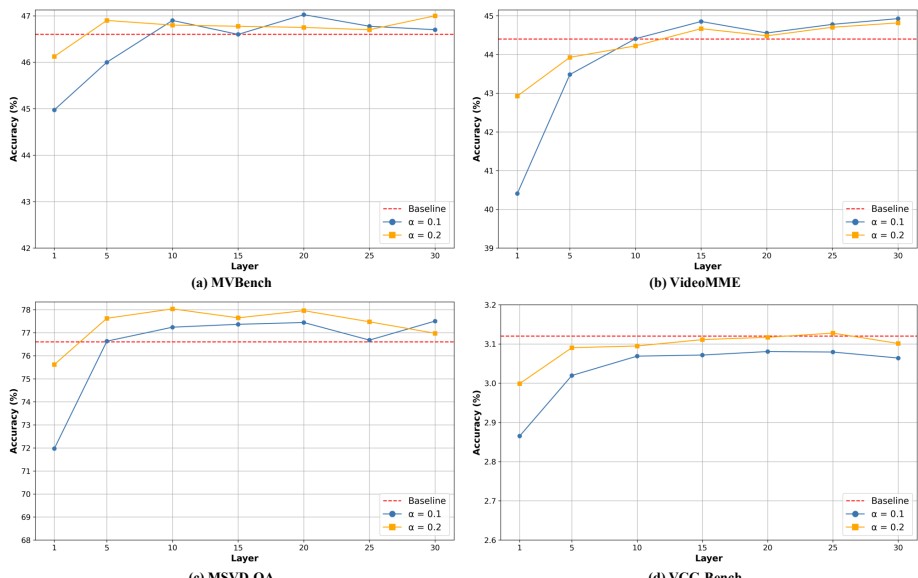

Figure 3: Ablation study on the effect of applying different retained ratios $\alpha$ and using attention weights from various attention layers on PLLaVA. We run each experiment three times to ensure stability.

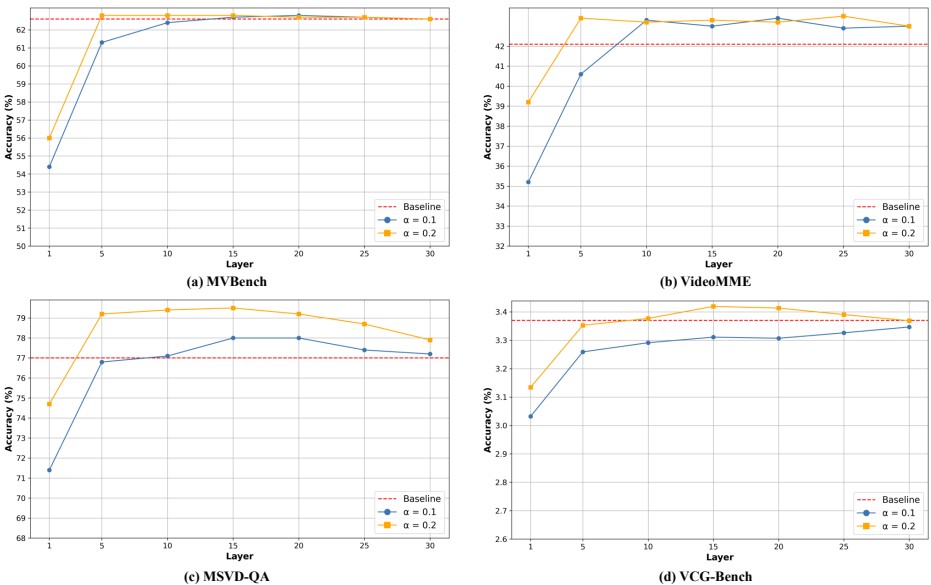

Figure 4: Ablation study on the effect of applying different retained ratios $\alpha$ and using attention weights from various attention layers on Tarsier. We run each experiment three times to ensure stability.

model's accuracy gradually saturates, allowing us to achieve performance comparable to the baseline across different models and datasets. This consistent trend indicates that we can apply similar parameter settings across various datasets and models without relying on extensive optimization and exploration.

**Qustion-to-Video Attention Visulization.** As shown in Fig. 5, we can observe that the highlighted attention areas across different layers are similar except for layer 1, which lacks focus on the informative regions. In contrast, layers 5, 10, and 15 effectively attend to the core objects. For instance,

**Question:** *Describe what the woman is performing.*

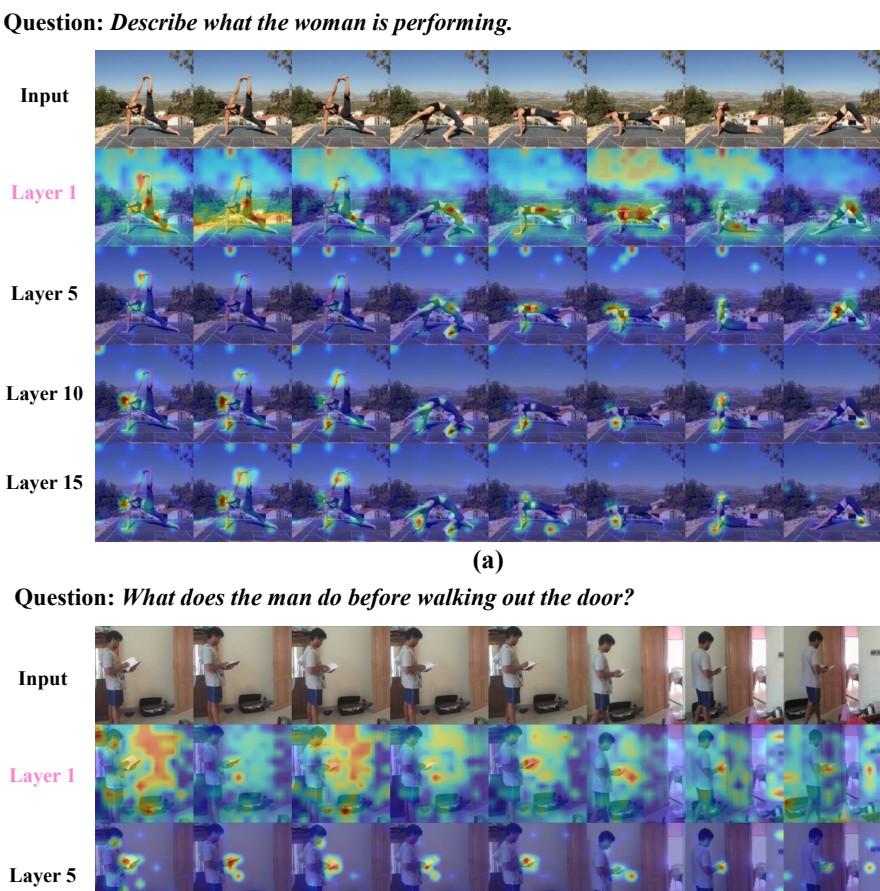

**(a)**

**Question:** *What does the man do before walking out the door?*

**(b)**

Figure 5: Visualization of question-to-video attention scores. We visualize the intermediate attention scores from the 1st, 5th, 15th, and 20th using the PLLaVA model. Best viewed with zooming in.

in Fig. 5 (a), key parts of the woman's exercise movements are emphasized, while in Fig. 5 (b), the focus is on the hand and book. These regions are relevant cues for answering the questions.

## 5 CONCLUSION

In this paper, we attempt to mitigate the significant computational overhead caused by video redundancy by leveraging the reasoning capabilities of LLMs. Specifically, we visualize the attention scores between question tokens and video frames, finding that question tokens effectively attend to visual signals relevant to the question content. Utilizing this intriguing observation, we design LLM-VTP, which leverages the ability of LLMs to identify useful visual signals for pruning visual features. We combine LLM-VTP with two video LLMs and conduct experiments on multiple video understanding tasks, thoroughly validating that this method can effectively reduce the number of tokens without compromising model performance.

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

# A APPENDIX

## A.1 QUANTITATIVE RESULTS

Table 4: Impacts of token merge.

| Method | MVBench |
|---|---|
| Ours | **46.9** |
| Ours *w*/token Merge | 46.7 |

Table 5: Impacts of the rank scope of token selection.

| Rank Scope | MVBench |
|---|---|
| Whole Sequence | 46.0 |
| Individual Frame | **46.9** |

Table 6: Impacts of pooling on attention heads.

| Method | MVBench |
|---|---|
| Max Pool | **46.9** |
| Mean Pool | 46.6 |
| Min Pool | 46.1 |

**Impacts of Token Merge.** Compared to other token pruning methods, PruMerge and Look-M, we do not merge the pruned tokens. To investigate the impact of design, we incorporate the pivotal merge strategy proposed by Look-M into our method, as shown in the Tab. 4. We find that this does not improve our model performance, which indicates that we have already identified the valuable tokens. In this scenario, merging the less informative tokens actually negatively impacts performance.

**Where to rank the tokens?** In Tab. 5, we investigate the impact of ranking token importance within different scopes. We compare ranking across the whole sequence with ranking within individual frames and find that the latter performs better. By analyzing the tokens selected by these two methods, we discover that due to the decay of position weights over distance (Su et al., 2024), the approach of ranking within the entire sequence tends to favor tokens from the last frame. This does not effectively reflect the relevance of the question tokens.

**Impacts of pooling methods applied on attention heads.** When handling multiple attention heads, we try different pooling methods, as shown in Tab. 6. We find that using max pooling yields the best results because it retains the most significant responses.

**Experiments on Long-form Benchmarks.** We conduct experiments on two popular long video datasets, VideoMME (Fu et al., 2024) and EgoSchema (Mangalam et al., 2023), and report the results in Tab. 7. For VideoMME, we follow the protocol that excludes the use of subtitles. Our experiments use two models, PLLaVA and Tarsier, and compare their accuracy against other methods. In our approach, we set the pruned layer $M$ to 10 and the retrained ratio $\alpha$ to 0.1 for both models. Similarly, the retrained ratio for Look-M is set to 0.1, while PruMerge+ employs its adaptive compression method, leading to an average retrained ratio of 0.36. The results indicate that our method most effectively preserves the original model's performance on these datasets compared to Look-M and PruMerge+, and in some cases, it even slightly enhances performance, demonstrating its effectiveness for long videos.

**Experiments with additional models.** Furthermore, in Tab. 8, we conduct experiments on two additional models, *i.e.,* VideoChat2 (Li et al., 2024) and ST-LLM (Liu et al., 2024a), and present performance comparisons on MVBench (Li et al., 2024), VideoMME (Fu et al., 2024), and Egoschema

Table 7: Performance comparison (top1-acc (%)) on the long-form datasets, *i.e.*, VideoMME (Fu et al., 2024) and Egochema (Mangalam et al., 2023).

| Method | Retained Ratio | VideoMME | | | | Egoschema | |
|---|---|---|---|---|---|---|---|
| | | Short | Medium | Long | Avg. | Subset | Fullset |
| Video-LLaVA (Lin et al., 2023) | 100% | 45.3 | 38.0 | 36.2 | 39.9 | - | - |
| Chat-UniVi-V1.5 (Jin et al., 2024) | 100% | 45.7 | 40.3 | 35.8 | 40.6 | - | - |
| mPLUG-Owl Ye et al. (2023) | 100% | - | - | - | - | - | 31.1 |
| InternVideo Wang et al. (2022) | 100% | - | - | - | - | - | 32.1 |
| **PLLaVA** | 100% | 52.2 | **44.4** | 36.5 | 44.4 | **47.8** | **42.6** |
| PLLaVA w/PruMerge+ | 36.1% | 51.3 | 43.7 | 35.1 | 43.4 | 44.5 | 40.7 |
| PLLaVA w/Look-M | 10.0% | 51.0 | 43.2 | 34.4 | 42.9 | 45.6 | 41.5 |
| **PLLaVA w/Ours** | 10.0% | **53.0** | 43.4 | **36.7** | **44.4** | 46.4 | 42.2 |
| **Tarsier** | 100% | 53.1 | 40.6 | 33.1 | 42.1 | 56.0 | **49.9** |
| Tarsier w/PruMerge+ | 35.9% | 51.4 | 39.8 | 36.1 | 42.4 | 52.8 | 47.3 |
| Tarsier w/Look-M | 10.0% | 51.1 | 37.6 | 31.1 | 39.9 | 55.5 | 46.4 |
| **Tarsier w/Ours** | 10.0% | **53.4** | **41.4** | 34.9 | **43.3** | **56.2** | 47.9 |

Table 8: Performance analysis (top1-acc (%)) using VideoChat2 (Li et al., 2024) and ST-LLM (Liu et al., 2024a)

| Method | FLOPs (T) | Retained Ratio | MVBench | VideoMME | | | | Egoschema | |
|---|---|---|---|---|---|---|---|---|---|
| | | | | Short | Medium | Long | Avg | Subset | Fullset |
| **VideoChat2** (Li et al., 2024) | 1.1 | 100% | 51.1 | 43.1 | 36.9 | 32.8 | 37.6 | 41.0 | 50.6 |
| VideoChat2 w/ Look-M | 1.1 | 10.0% | 50.0 | 41.9 | 35.0 | 32.6 | 36.5 | 40.9 | 50.7 |
| **VideoChat2 w/ Ours** | 0.5 | 10.0% | **51.3** | **43.4** | 36.2 | **34.6** | **38.1** | **41.8** | **52.6** |
| **ST-LLM** (Li et al., 2024) | 5.4 | 100% | 54.9 | **51.4** | 39.7 | 34.8 | **42.0** | 45.6 | **56.2** |
| ST-LLM w/ Look-M | 5.4 | 10.0% | 54.0 | 50.6 | 38.0 | 33.1 | 40.6 | 44.5 | 54.4 |
| **ST-LLM w/ Ours** | 2.1 | 10.0% | **54.9** | 51.3 | **39.7** | **34.8** | 41.9 | **45.6** | 56.0 |

(Mangalam et al., 2023). These comparisons highlight our model's effectiveness for both long and short videos. Consistent with the established approach in PLLaVA, we perform token pruning at Layer 10 with a prune ratio of 0.1 and apply the same prune ratio for Look-M. Since Llava-PrugMerge is incompatible with the Q-Former architecture for token compression, it is excluded from our comparison. We find that using the same hyperparameters as PLLaVA, our model better maintains the original baseline accuracy compared to Look-M, and in some cases, it even enhances performance. For instance, with VideoChat2, we observe performance improvements in nearly all scenarios. These experiments further demonstrate that our hyperparameter settings can be effectively transferred across different models and benchmarks. Besides, our method requires much lower computational cost than Look-M and the baseline, which validates its efficiency.

**Assessing Baseline Maintenance of Token Pruning Methods.** To comprehensively assess how well different token pruning methods maintain the original baseline performance, we calculate the accuracy ratios of each method relative to the baseline across various datasets, and average the results across PLLava and Tarsier. The results are reported in Tab. 9. Our method consistently approaches the baseline performance more closely in every scenario, demonstrating its stability.

**Assessing Performance on samples requiring advanced reasoning capacity.** To further validate the "LLM-Reasoned" capability, we conduct experiments on the MVBench dataset using the PLLaVA model, selecting categories that require strong LLM reasoning abilities, including Action Sequence, Action Prediction, Episodic Reasoning, and Counterfactual Inference. We compare the following approaches: (a) The baseline method where all tokens are utilized for testing. (b) Our proposed token pruning method leverages LLM reasoning to select the most relevant tokens to answer the question. (c) Utilizing the tokens that are pruned by our method. As shown in Tab. 10, our method effectively maintains the baseline accuracy (even improves in some cases), indicating that it effectively identifies the most critical information necessary for answering the questions. In contrast, when using the pruned tokens, the model's accuracy significantly decreases, despite processing a much larger set of tokens. These results demonstrate that our method effectively harnesses the LLM's reasoning capabilities to focus on relevant content and disregard irrelevant information. This corroborates the effectiveness our "LLM-Reasoned" token pruning method, as the model relies on reasoning to select vital tokens that are not immediately apparent from the question alone.

Table 9: Comparison of accuracy ratios between token pruning methods over the baselines, averaged across PLLava and Tarsier.

| Method | VCG-Bench (Avg) | MSVD (Acc) | MSRVTT (Acc) | ActivityNet (Acc) | TGIF (Acc) | Average |
|---|---|---|---|---|---|---|
| Prumerge | 98.3% | 100.5% | 96.7% | 99.4% | 99.0% | 98.8% |
| Look-M | 90.1% | 101.0% | 99.2% | 97.2% | 99.0% | 97.4% |
| Ours | **98.5%** | **102.1%** | **100.3%** | **99.5%** | **99.3%** | **99.9%** |

Table 10: Performance Comparison (top-1 ACC (%)) on Samples Requiring Advanced Reasoning Capabilities with the PLLaVA Model.

| Method | Retained Ratio | Action Sequence | Action Prediction | Episodic Reasoning | Avg. |
|---|---|---|---|---|---|
| All Tokens | 100% | 58.0 | 49.0 | 48.0 | 51.7 |
| Ours | 20% | **60.0** (+2.0) | **51.0** (+2.0) | **48.5** (+0.5) | **53.2** (+1.5) |
| Pruned Tokens | 80% | 50.5 (-7.5) | 46.0 (-3.0) | 46.5 (-1.5) | 47.7 (-4.0) |

## A.2 QUALITATIVE RESULTS

**Side-by-side Visualizations of Attention Maps and Token Selection.** In Fig. 6, we present a side-by-side comparison demonstrating how our model selects tokens guided by attention scores, highlighting the LLM's strength in focusing on informative regions related to the questions.

Below, we provide detailed analyses on how our model leverages the *reasoning capabilities of LLMs* to locate relevant visual regions that are not explicitly mentioned in the questions:

(1) Figure 6 (a): The model identifies the key object mentioned in the question, the *blanket*, and uses the temporal cue (*after*) from the question to locate additional relevant visual elements, such as objects on the table and the person's hand movements in frames 2, 6, and 7. These details are not directly provided in the question and are inferred through the model's reasoning abilities.

(2) Figure 6 (b): The model accurately detects the action of the person holding food in frames 7 and 8, and infers that the presence of a *bag* the person puts down is relevant for answering the question, even though the *bag* is not mentioned. This demonstrates the model's ability to reason about relevant objects based on contextual cues.

(3) Figure 6 (c): Despite the absence of any mention of a *book* in the question, the model correctly identifies critical visual regions related to the *book* by reasoning over the visual content and context provided.

(4) Figure 6 (d): The model focuses on the person's hand movements, which are crucial for answering the question. Even though the question does not emphasize hand motions, the model infers the importance of these actions through reasoning.

These examples showcase how our model utilizes LLM reasoning to identify and focus on pertinent visual information that is not explicitly described in the questions.

**Attention Map Comparison.** In Fig. 7 and Fig. 8, we include comparisons between our LLM's attention maps and those of several strong video encoders, including UMT (Li et al., 2023b), ActionCLIP (Wang et al., 2021), and InternVideo2 (Wang et al., 2024b). The results show that, unlike these models, the question-to-vision attentions in the LLM accurately focus on visual tokens that are pertinent to the question. In contrast, the other models often struggle to pinpoint key tokens and may focus on irrelevant objects or background elements. These observations suggest that LLMs possess a unique ability to align visual information with linguistic context through their reasoning capabilities, which is not simply a byproduct of standard attention mechanisms in typical video encoders.

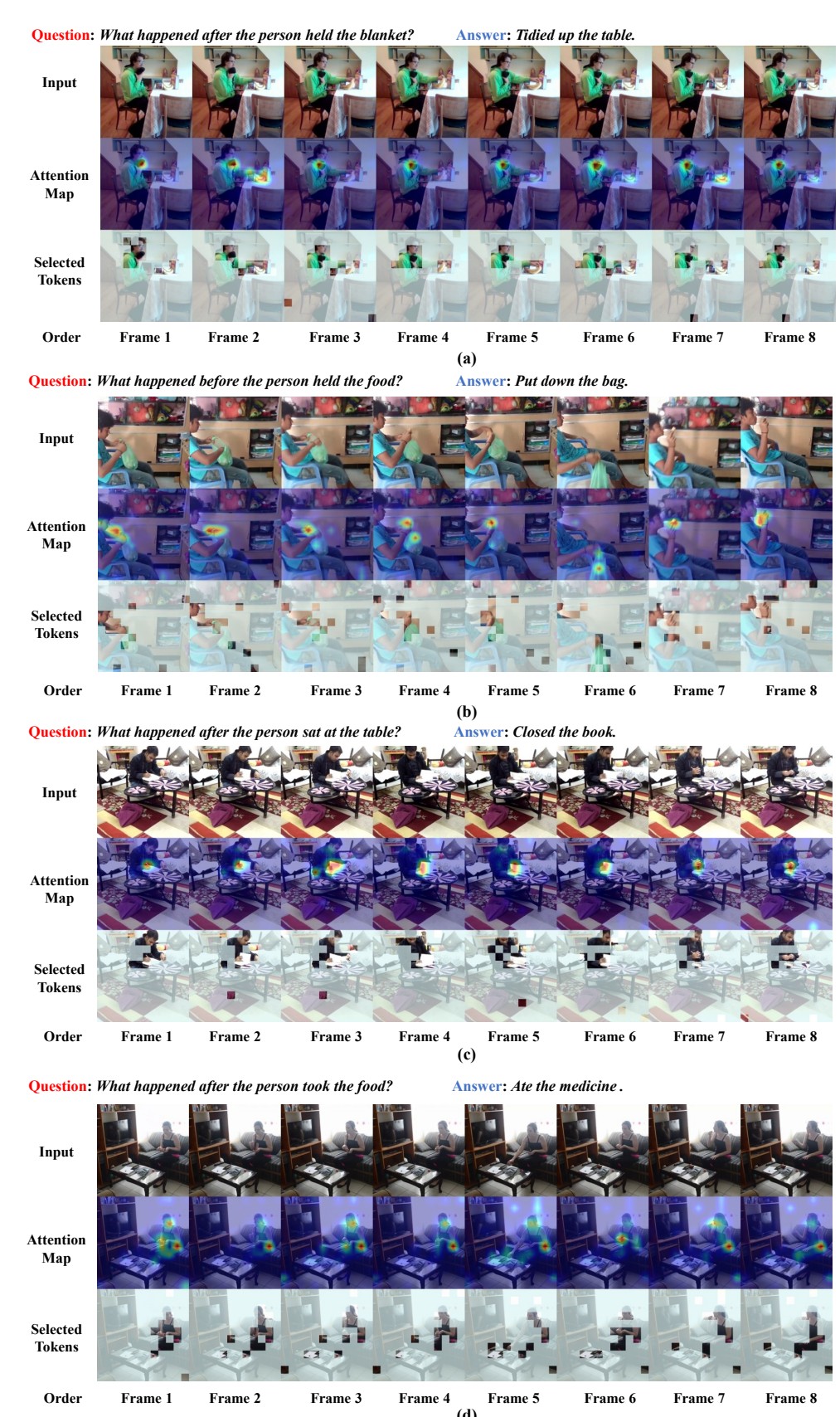

Figure 6: Visualization of the question-to-visual attentions and token selection of LLM-VTP.

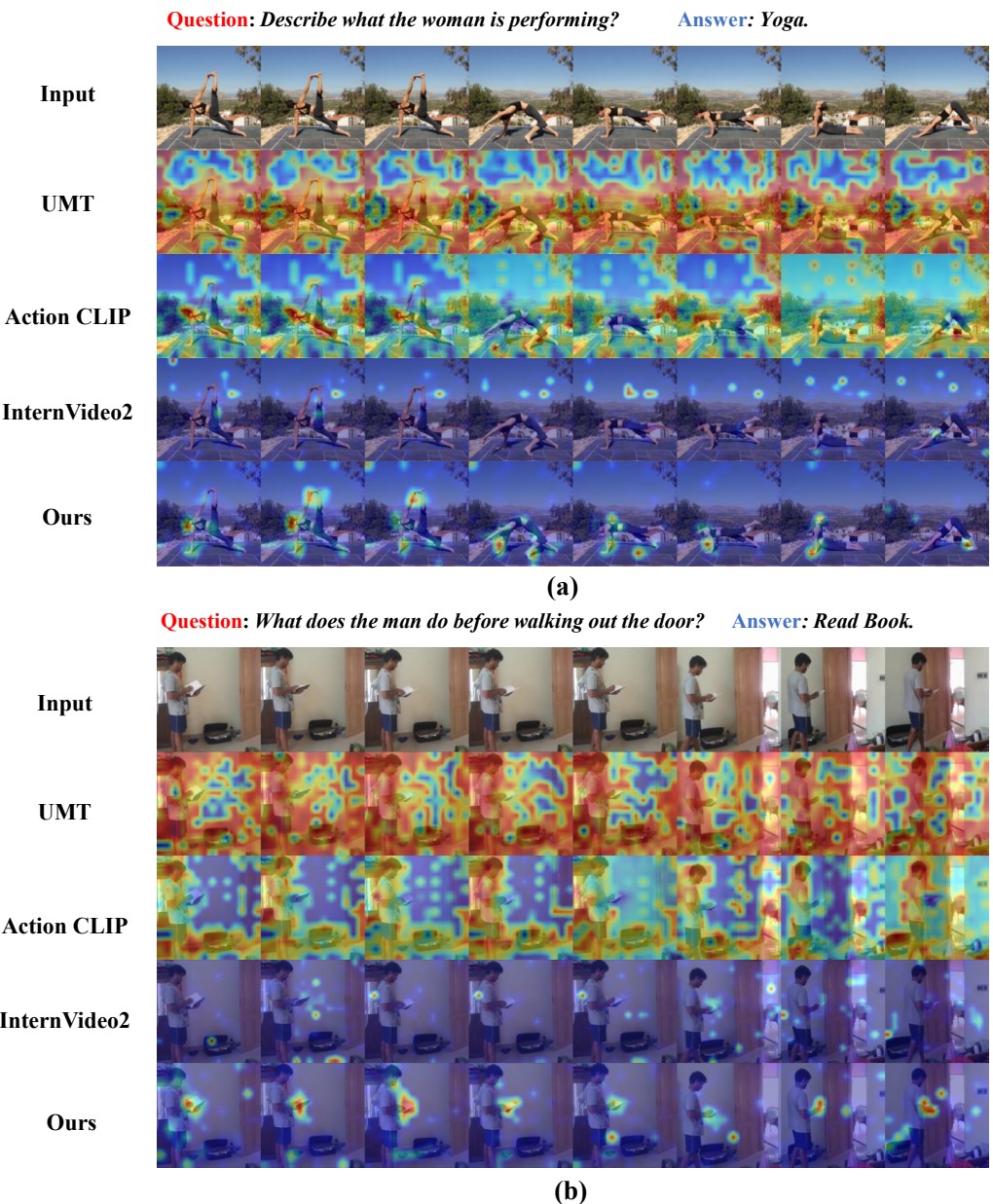

Figure 7: Attention map Comparison of video encoders and our method.

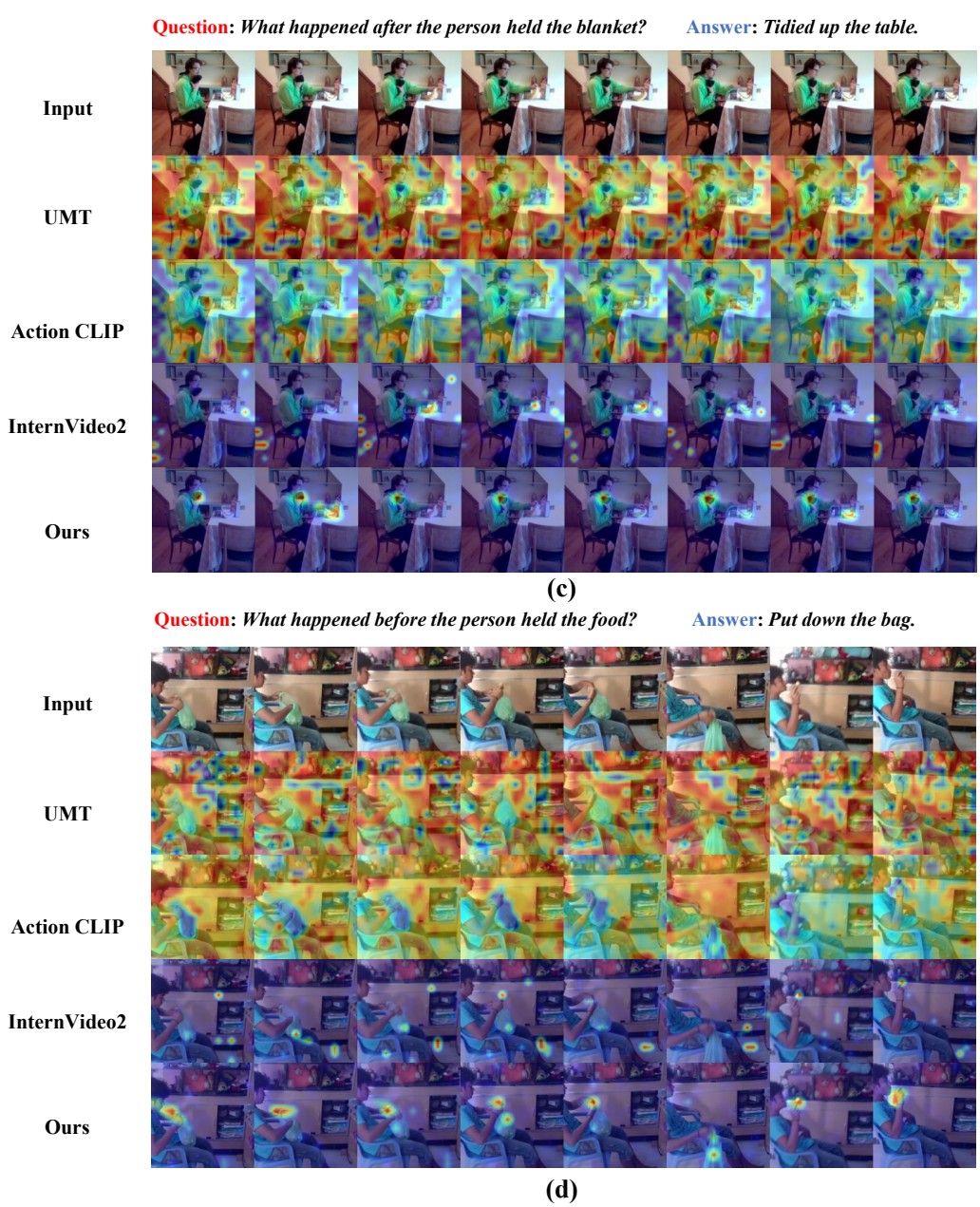

Figure 8: Attention map Comparison of video encoders and our method.

