# OpenReview forum: "LLM-VTP: LLM-Reasoned Visual Token Pruning for Efficient Multi-Modal Video Understanding"
_ICLR.cc/2025/Conference — Submitted to ICLR 2025_

### Official Review · Reviewer_iKaN · 2024-10-26

**Soundness:** 3
**Presentation:** 2
**Contribution:** 2
**Rating:** 5
**Confidence:** 3

**Summary:**

This paper proposes a video LLM that focuses on visual token pruning. While previous work has used training-based approaches to reduce redundancy in video data, the authors consider them suboptimal due to the lack of question-specific designs. They develop a training-free pruning method based on attention scores between questions and video tokens. In particular, only the top-ranked video tokens are retained and the KV cache is compressed accordingly, which significantly reduces the computational cost. The proposed method is validated with two video LLMs on multiple video understanding benchmarks, showing significant speedup by pruning 80-90% of the tokens with competitive performance.

**Strengths:**

* The paper is generally well written and easy to follow.
* The authors' analysis of the current limitations of visual token pruning approaches is reasonable, and the proposed approach directly addresses these limitations.
* Extensive quantitative and qualitative results are provided to support the effectiveness of the proposed method.

**Weaknesses:**

* The "LLM-reasoned" in the tile is an overclaim. While LLMs have demonstrated impressive reasoning capabilities, they are not explicitly utilized in the paper. There is also a lack of evidence or further analysis to prove that the LLM has implicitly reasoned about how to prune the visual tokens. The visualizations in Figure 1b and Figure 4 may be the authors' attempts to illustrate this, but they seem too vague for me to draw any conclusions.
* Limited technical contribution. Both token pruning and KV compression are widely used techniques in (multimodal) LLMs, and the proposed approach through attention score-based top-$\alpha\%$ selection does not seem to be a significant contribution. This strategy should be sensitive to the hyperparameter $\alpha$ and might fail on certain videos with more dense information.
* The comparative results with PruMerge and Look-M in Table 2 are mixed. The proposed method does not always achieve a better performance-efficiency tradeoff than the baselines, and there is no intuitive summary of the comparison. The authors have provided some explanation in lines 362-371, but they do not fully address these concerns.
* The retained ratio may not be not a good metric because it does not consider token merging as an effective strategy for reducing computational cost. For token merging, the authors only mentioned that "merging the less informative tokens actually negatively impacts performance" in line 485, which is not a very solid argument since the result is only obtained from a single comparison. It would be better if additional metrics such as FLOPS are included to provide a comprehensive analysis of computational efficiency.

**Questions:**

* Could the authors provide visualizations of which tokens are pruned by their ranking-based selection strategy? Perhaps in the form of a side-by-side comparison of the original and pruned attention heatmaps. This would make the results more intuitive.
* Could the authors tune the hyperparameters for the PruMerge baseline and compare with it on the same retained ratio? I cannot tell if the performance disadvantage of the proposed method is due to a higher compression rate or some weakness of the method.

---

> ### Author Response · Authors · 2024-11-22
> **Response to Reviewer iKaN (1/4)**
>
> Thank you for taking the time to review our paper and for providing detailed and constructive feedback. We are pleased to hear that you found the paper to be well-written and easy to follow and that you consider the proposed method to be reasonable.
>
> Please refer to the detailed response below regarding the raised several points.
>
> >**Q1:** The "LLM-reasoned" in the tile is an overclaim. While LLMs have demonstrated impressive reasoning capabilities, they are not explicitly utilized in the paper. There is also a lack of evidence or further analysis to prove that the LLM has implicitly reasoned about how to prune the visual tokens. The visualizations in Figure 1b and Figure 4 may be the authors' attempts to illustrate this, but they seem too vague for me to draw any conclusions.
>
> **R:** In our paper, the term “LLM-reasoned” does not imply that the model provides explicit outputs for guiding token pruning. Rather, it refers to leveraging the latent knowledge embedded in the intermediate layers of the LLM to identify informative regions. This information is then used to guide the model in pruning irrelevant visual tokens. Additionally, our approach of utilizing intermediate layer knowledge, as opposed to relying on explicit outputs, reduces computational overhead. It allows the model to identify key visual features during the forward pass through the LLM layers, eliminating the need to wait until the final output layer to make these determinations.
>
> Moreover, if you have any suggestions for modifying the title to make it more appropriate, we would greatly appreciate your input.
>
> >**Q2:** Limited technical contribution. Both token pruning and KV compression are widely used techniques in (multimodal) LLMs, and the proposed approach through attention score-based top-α selection does not seem to be a significant contribution. This strategy should be sensitive to the hyperparameter alpha and might fail on certain videos with more dense information.
>
> **R:** In multimodal LLM inference, improving inference speed in long-context scenarios is a critical technical challenge. Within multimedia data, video is a mainstream format, with its understanding intrinsically dependent on long-context inference. Therefore, investigating video inference acceleration is essential.
>
> We are the first to predominantly focus on exploring token pruning for video LLMs. While our method is simple, we believe that its simplicity is a merit rather than a drawback. To assess the sensitivity of our approach to the hyper-parameters, we provide additional experiments across various datasets (Multi-choice benchmarks: MVBench and VideoMME, Open-ended Benchmarks: VCGBench and MSVD-QA) and models (PLLaVA and Tarsier), as shown in Figures 3 and 4 of the main paper. The results indicate that starting from layer 5, the model’s accuracy gradually saturates across both models and all benchmarks. This allows us to achieve performance comparable to the baseline across various models and datasets for both $\alpha$=0.1 and $\alpha$=0.2. **This consistent trend suggests that similar parameter settings can be applied across different datasets and models without requiring extensive optimization and exploration.**
>
> (The response to this question continues to the next comment (2/4))

---

> ### Author Response · Authors · 2024-11-22
> **Response to Reviewer iKaN (2/4)**
>
> (Continued to Q2)
>
> Furthermore, in Table 1 below, we conduct experiments on two additional models, VideoChat2 [1] and ST-LLM [2], and present performance and computational cost comparisons on MVBench, VideoMME, and Egoschema. These comparisons highlight our model’s effectiveness for both long and short videos. **Consistent with the established approach in PLLaVA, we perform token pruning at Layer 10 with a retained ratio ($\alpha$) of 0.1 (90\% tokens are pruned)** and apply the same prune ratio for Look-M. Since Llava-PrugMerge is incompatible with the Q-Former architecture for token compression, it is excluded from our comparison. We find that using the same hyperparameters as PLLaVA, our model better maintains the original baseline accuracy compared to Look-M, and in some cases, it even enhances performance. **These experiments further demonstrate that our hyperparameter settings can be effectively transferred across different models and benchmarks.** Besides, we achieve much lower computational cost than Look-M and the baseline, which demonstrates our efficiency.
>
> **Table 1: Performance analysis (top1-acc (\%)) using VideoChat2 and ST-LLM.**
> |      Method          | FLOPs (T) | MVBench | VideoMME (Short) | VideoMME (Medium) | VideoMME (Long) | VideoMME (Avg) | Egoschema (Full-set) | Egoschema (Sub-set) |
> |-----------------------|:---------:|:------------------:|:-------------------:|:-----------------:|:----------------:|:-----------------------:|:----------------------:|:----------------:|
> | VideoChat2   |   1.1       | 51.1    | 43.1             | **36.9**          | 32.8            | 37.6           | 41.0                  | 50.6                 |
> | VideoChat2 w/ Look-M | 1.1  | 50.0    | 41.9             | 35.0              | 32.6            | 36.5           | 40.9                  | 50.7                 |
> | **VideoChat2 w/ Ours** | **0.5**   | **51.3**| **43.4**         | 36.2              | **34.6**        | **38.1**       | **41.8**              | **52.6**             |
> |                       |         |                  |                   |                 |                |                       |                      |
> | ST-LLM           |  5.4    | **54.9**| **51.4**         | **39.7**          | **34.8**        | **42.0**       | **45.6**              | **56.2**             |
> | ST-LLM w/ Look-M  | 5.4    | 54.0    | 50.6             | 38.0              | 33.1            | 40.6           | 44.5                  | 54.4                 |
> | **ST-LLM w/ Ours**     | **2.1**   | **54.9**   | 51.3             | **39.7**              | **34.8**            | 41.9           | **45.6**                  | 56.0                 |
>
> To assess the model’s performance on videos with denser information, we select content-rich videos from MVBench. Specifically, we use the CLIP-VIT-B/16 model to process video frames and calculate the average patch-feature similarity (*i.e.*, cosine distance) between adjacent frames. Video samples with low similarity between adjacent frames are considered to have more temporal variation, less redundancy, and higher information content. As shown in Table 2 below, we test the top 10% most informative samples (*i.e.*, 460 video-question pairs in total) on MVBench and find that our method maintains baseline performance on these challenging samples, even showing a slight improvement of +0.2%. This indicates that our model remains effective on high-information video samples.
>
> **Table 2: Performance analysis (top1-acc (\%)) of dense-information samples on MVBench.**
> | Method         | All Samples | Top-10% Dense-Information Samples |
> |----------------|:-------------:|:---------------------------------:|
> | PLLaVA          | 46.6        | 43.7                            |
> | PLLaVA w/Ours  | **46.9**        | **43.9**                            |

---

> ### Author Response · Authors · 2024-11-22
> **Response to Reviewer iKaN (3/4)**
>
> >**Q3:** The comparative results with PruMerge and Look-M in Table 2 are mixed. The proposed method does not always achieve a better performance-efficiency tradeoff than the baselines, and there is no intuitive summary of the comparison. The authors have provided some explanation in lines 362-371, but they do not fully address these concerns.
>
> **R:** To comprehensively assess how well different token pruning methods maintain the original baseline performance, we calculate the accuracy ratios (%) of each method relative to the baseline, and average the results across PLLava and Tarsier. In Table 3 below, we find that our method consistently approaches baseline performance more closely (some even better) in every scenario, demonstrating its reliable effectiveness.
>
> We have included this table and the corresponding discussion in Appendix A.1 of the revised paper.
>
> **Table 3: Comparison of accuracy ratios between token pruning methods over the baselines, averaged across PLLava and Tarsier.**
> | Method   | Video-ChatGPT-Bench (Avg) | MSVD (Acc) | MSRVTT (Acc) | ActivityNet (Acc) | TGIF (Acc) | Average |
> |----------|:---------------------------:|:------------:|:--------------:|:------------------:|:-----------:|:---------:|
> | Prumerge | 98.3%                     | 100.5%     | 96.7%        | 99.4%            | 99.0%     | 98.8%   |
> | Look-M   | 90.1%                     | 101.0%     | 99.2%        | 97.2%            | 99.0%     | 97.4%   |
> | Ours     | **98.5%**                 | **102.1%** | **100.3%**   | **99.5%**        | **99.3%** | **99.9%** |
>
> > **Q4:** The retained ratio may not be not a good metric because it does not consider token merging as an effective strategy for reducing computational cost. For token merging, the authors only mentioned that "merging the less informative tokens actually negatively impacts performance" in line 485, which is not a very solid argument since the result is only obtained from a single comparison. It would be better if additional metrics such as FLOPS are included to provide a comprehensive analysis of computational efficiency.
>
> **R:** Thank you for your constructive feedback. As shown in Table 4 below, we evaluate the average computational cost of LLMs using different pruning methods on the VideoChatGPT-Bench dataset for text generation. Our approach demonstrates lower computational cost. While Look-M has a computational cost similar to the baseline, its slow inference speed is due to its reliance on vanilla attention for every LLM layer, which significantly hampers efficiency. Additionally, although PruMerge+ achieves comparable FLOPs to ours, it relies on the [CLS] token for token compression, which is incompatible with some video LLMs based on adapted projectors (*e.g.,* ST-LLM and VideoChat2 with Q-former architecture), thus limiting its applicability.
>
> The FLOPs comparison has been included in Table 3 of our revised paper.
>
> **Table 4: Efficiency comparison of different pruning methods on VideoChatGPT Bench.**
> | Method                | Average Decoding Length | Inference Time (s) | Speed Up | FLOPs (T) |
> |-----------------------|:-------------------------:|:--------------------:|:----------:|:-------:|
> | PLLaVA                | 72.5                    | 3.12               | -        | 26.8  |
> | PLLaVA w/PruMerge+    | 72.0                    | 2.67               | 1.2×     | 9.2   |
> | PLLaVA w/Look-M       | 72.0                    | 6.04               | 0.5×     | 26.7  |
> | PLLaVA w/LLM-VPT (Ours)| **73.0**               | **2.24**           | **1.4×**     | **8.8**  |

---

> ### Author Response · Authors · 2024-11-22
> **Response to Reviewer iKaN (4/4)**
>
> > **Q5:** Could the authors provide visualizations of which tokens are pruned by their ranking-based selection strategy? Perhaps in the form of a side-by-side comparison of the original and pruned attention heatmaps. This would make the results more intuitive.
>
> **R:** Thanks for your valuable suggestion. We have included a side-by-side comparison in Figure 6 of the Appendix. We can find that the model will select informative tokens aligned with the activations of attention maps.
>
> > **Q6:** Could the authors tune the hyperparameters for the PruMerge baseline and compare with it on the same retained ratio? I cannot tell if the performance disadvantage of the proposed method is due to a higher compression rate or some weakness of the method.
>
> **R:** The PruMerge method employs an adaptive pruning method, which means its pruning ratio cannot be manually adjusted to match our method. To evaluate the impact of the pruning rate on the results, we adjust our method to align with PruMerge’s pruning ratio in Table 5 below. We observe improvements in our model’s performance on VCG-Bench, which demonstrates that a higher retained ratio does not harm the performance. The unsatisfactory performance of PruMerge stems from its reliance on vision-only features to prune tokens without considering the context of the questions. This approach may result in the loss of critical visual cues needed for accurately answering the questions.
>
>
> **Table 5: Performance comparison by aligning our retained ratio to Prumerge+. \* indicates using the same retained ratio as Prumerge.**
> |       Method       | MVBench | VCG-Bench CI | VCG-Bench DO | VCG-Bench CU | VCG-Bench TU | VCG-Bench CO | VCG-Bench Avg. |
> |---------------------|:---------:|:--------------:|:--------------:|:--------------:|:--------------:|:--------------:|:----------------:|
> | PLLaVA              | 46.6    | **3.21**     | **2.86**     | **3.62**     | 2.33         | 2.93         | **3.12**       |
> | PLLaVA w/ Prumerge | 45.9    | 3.17         | 2.78         | 3.50         | 2.37         | 2.91         | 3.07           |
> | PLLaVA w/ Ours      | **46.9**    | 3.16         | 2.80         | 3.48         | 2.39         | 2.91         | 3.08           |
> | PLLaVA w/ Ours*     | **46.9**| 3.20         | 2.80         | 3.54         | **2.44**     | **3.01**     | 3.11           |
>
> Thanks again for your valuable comments and looking forward to your reply.

---

> ### Comment · Reviewer_iKaN · 2024-11-24
>
> Thank you for conducting the additional comparisons. However, I am still not satisfied with the explanation on the overclaim of "LLM-reasoned", there should be some empirical evidence to justify it.

---

> ### Author Response · Authors · 2024-11-25
> **Response to Reviewer iKaN Comment**
>
> > Thank you for conducting the additional comparisons. However, I am still not satisfied with the explanation on the overclaim of "LLM-reasoned", there should be some empirical evidence to justify it.
>
> **R:**  Thank you for your thoughtful feedback and for acknowledging our additional experimental results. To address the concern regarding "**LLM-reasoned**", we conduct additional analyses and experiments **qualitatively** and **quantitatively**. We have also **revised our paper** to reflect them.
>
> (1) **Qualitative Analysis**
>
> Please refer to Figure 6 in Appendix A.2 for detailed visualizations. The following explanation illustrates **how our model leverages the reasoning capabilities of LLMs to locate relevant visual regions that are not explicitly mentioned in the questions:**
>
> - **Figure 6(a):** The model identifies the key object mentioned in the question, the *blanket*, and uses the temporal cue (*after*) from the question to locate additional relevant visual elements, such as objects on the table and the person's hand movements in frames 2, 6, and 7. These details are not directly provided in the question and are inferred through the model's reasoning abilities.
>
> - **Figure 6(b):** The model accurately detects the action of the person holding food in frames 7 and 8, and infers that the presence of a *bag* the person puts down is relevant for answering the question, even though the bag is not mentioned. This demonstrates the model's ability to reason about relevant objects based on contextual cues.
>
> - **Figure 6 \(c\):** Despite the absence of any mention of a *book* in the question, the model correctly identifies critical visual regions related to the book by reasoning over the visual content and context provided.
>
> - **Figure 6(d):** The model focuses on the person's hand movements, which are crucial for answering the question. Even though the question does not emphasize hand motions, the model infers the importance of these actions through reasoning.
>
> These examples showcase how our model utilizes LLM reasoning to **identify and focus on pertinent visual information that is not explicitly described in the questions**.
>
> (2) **Quantitative Analysis**
>
> To further support our claim, we conducted experiments on the MVBench dataset using the PLLaVA model, focusing on categories that demand advanced LLM reasoning abilities—namely, Action Sequence, Action Prediction, and Episodic Reasoning. In these categories, answering the questions requires the model to leverage its reasoning capabilities to identify the relevant visual tokens.
>
> We compare the following approaches: (a) The baseline method where all tokens are utilized for testing. \(b\) Our proposed token pruning method leverages LLM reasoning to select the most relevant tokens for answering the question. \(c\) Utilizing the tokens that are pruned by our method.
>
> **Table 1. Performance Comparison on Samples Requiring Advanced Reasoning Capabilities with the PLLaVA Model.**
> | Method         | Retained Ratio | Action Sequence   | Action Prediction   | Episodic Reasoning | Avg.            |
> |----------------|----------------|-------------------|---------------------|-------------------|-----------------|
> | All Tokens     | 100\%          | 58.0              | 49.0                | 48.0          | 51.7            |
> | Ours           | 20\%           | **60.0** (+2.0)   | **51.0** (+2.0)     | **48.5** (+0.5)       | **53.2** (+1.5) |
> | Pruned Tokens  | 80\%           | 50.5 (-7.5)       | 46.0 (-3.0)         | 46.5 (-1.5)       | 47.7 (-4.0)     |
>
> Our method effectively maintains baseline accuracy (even improves under some cases) indicating that it effectively identifies the critical information necessary for answering the questions. In contrast, when using the pruned tokens, the model’s accuracy significantly decreases, despite processing a much larger set of tokens (4 times).
>
> These results demonstrate that our method effectively harnesses the LLM's reasoning capabilities to focus on relevant content and disregard irrelevant information. This supports our idea of “LLM-reasoned” token pruning, where the model uses reasoning to choose important tokens that aren’t obvious from the question.
>
> We have revised our paper to include the above analysis, experiments, and visualizations in **Appendix A.2**. Please check it out.
>
> We hope that the additional qualitative and quantitative evidence provided could address your concerns and justify our use of the term "**LLM-reasoned**" in describing our token pruning method. Meanwhile, we are also open to suggestions for other alternatives.
>
> We sincierly hope that our response above can address the reviewer's concern and looking forwards to further discussion if there is any remaining concern.

---

> ### Comment · Reviewer_iKaN · 2024-11-25
>
> Thank you for the comprehensive reply and for visualizing the inner process of multimodal LLMs. However, I don't think the attention map can be considered reasoning, since any video encoder could have this kind of attention map. Also, it is very unclear whether LLM are capable of complex spatial reasoning on attention maps, given that they have only shown extensive world knowledge and verbal reasoning abilities.

---

> ### Author Response · Authors · 2024-11-27
> **Response to Reviewer iKaN Comment**
>
> Thanks for your feedback. Below, we provide additional visualizations, analyses, and clarifications to address your concerns.
>
> > Q1:  However, I don't think the attention map can be considered reasoning since any video encoder could have this kind of attention map.
>
> **R:** We appreciate the thoughtful feedback and would like to clarify a potential misunderstanding here. In our paper, we do not consider the attention map itself as the reasoning process. Instead, we leverage the inherent reasoning capabilities of LLMs to discover which visual tokens are critical for answering a given question. The attention map serves as an illustration of this reasoning ability, indicating what tokens the model focuses on, rather than being the mechanism by which reasoning occurs.
>
> In both the main text and the rebuttal, we provide qualitative and quantitative evidence demonstrating that attention maps can effectively showcase the LLM’s reasoning ability. These maps reveal that the LLM can identify and focus on the most pertinent visual tokens related to the question, allowing us to efficiently prune up to 80-90% of irrelevant tokens while maintaining the performance.
>
> Furthermore, the reasoning ability exhibited by LLMs, as reflected in their attention maps, is not a characteristic inherently shared by all video encoders. To illustrate this point, in Fig. 7 and Fig. 8 in Appendix A.2, we include comparisons between our LLM’s attention maps and those of several strong video encoders, including UMT [1], ActionCLIP [2], and InternVideo2 [3]. The results show that, unlike these models, the attentions in the LLM accurately focus on visual tokens that are pertinent to the question. In contrast, the other models often struggle to pinpoint key tokens and may focus on irrelevant objects or background elements.
>
> These observations suggest that LLMs possess a unique ability to align visual information with linguistic context through their reasoning capabilities, which is not simply a byproduct of standard attention mechanisms in typical video encoders.
>
> > Q2: Also, it is very unclear whether LLM are capable of complex spatial reasoning on attention maps, given that they have only shown extensive world knowledge and verbal reasoning abilities.
>
> **R:** Thank you for bringing up this important point. Similar to our response to the above comment, we would like to emphasize that the LLM does not perform reasoning on the attention maps; rather, the attention maps are used to illustrate the LLM’s reasoning processes. The maps provide insights into how the model leverages its reasoning abilities to focus on relevant visual tokens in response to a question.
>
> Regarding the capability of LLMs to perform complex spatial reasoning, we acknowledge that this is an area that extends beyond the study scope of our paper. Our work aims to utilize the innate capacities of video LLMs to enhance inference efficiency by pruning irrelevant visual tokens, rather than to provide an in-depth exploration of their spatial reasoning abilities.
>
> That said, we believe that LLMs do possess some degree of spatial reasoning capability. This is supported by recent advancements and architectural designs in multimodal models. For instance, designs such as the “Any-resolution” (AnyRes) approach in LLaVA-NeXt [4] and the use of Multimodal Rotary Positional Embeddings (mRoPE) in Qwen-2-VL [5] rely on the model’s ability to process and reason about spatial information. These methods involve rearranging images, integrating multiple patches per image, and implementing sophisticated positional encodings, which assume that the underlying LLM can interpret and reason with spatial relationships.
>
> Moreover, our experimental results provide additional evidence of the LLM’s spatial reasoning abilities. Despite pruning up to 90% of the visual tokens, our proposed method retains the model’s performance on benchmarks that require understanding temporal and spatial relationships within videos. This would be unlikely if the LLM lacked the capacity to reason about spatial information, as the pruned tokens would greatly impact the model’s understanding in the absence of such reasoning.
>
> We also note that current video understanding benchmarks primarily focus on temporal-related reasoning, rather than explicitly testing complex spatial reasoning. Therefore, while our work demonstrates the model’s effectiveness in these domains, a comprehensive evaluation of complex spatial reasoning capabilities would require dedicated benchmarks designed for that purpose, which are currently lacking in the video understanding community.
>
> We sincerely hope that our response above can address the reviewer's concern and look forward to further discussion if there is any remaining concern.

---

> > ### Author Response · Authors · 2024-11-27
> > **References**
> >
> > **References**
> >
> > [1] Li, Kunchang, et al. "Unmasked teacher: Towards training-efficient video foundation models." Proceedings of the IEEE/CVF International Conference on Computer Vision. 2023.
> >
> > [2] Wang, Mengmeng, Jiazheng Xing, and Yong Liu. "Actionclip: A new paradigm for video action recognition." arXiv preprint arXiv:2109.08472 (2021).
> >
> > [3] Wang, Yi, et al. "Internvideo2: Scaling video foundation models for multimodal video understanding." arXiv e-prints (2024): arXiv-2403.
> >
> > [4] Liu, Haotian, et al. "Llava-next: Improved reasoning, ocr, and world knowledge." Jan. 2024,
> >
> > [5] Wang, Peng, et al. "Qwen2-vl: Enhancing vision-language model's perception of the world at any resolution." arXiv preprint arXiv:2409.12191 (2024).

---

> ### Comment · Reviewer_iKaN · 2024-11-27
>
> Thank you for the detailed answer. After reading all your responses, I am still concerned that this key idea (pruning by reasoning) is an overclaim, therefore I would like to maintain my score.

---

> > ### Author Response · Authors · 2024-11-28
> > **Response to Reviewer iKaN Comment**
> >
> > > Q: After reading all your responses, I am still concerned that this key idea (pruning by reasoning) is an overclaim.
> >
> > **R:** Thank you for your response. In this rebuttal discussion, we have provided thorough answers to the concerns regarding the novelty of our work, the experimental results, efficiency comparisons, visualizations, and the in-depth ablation experiments. We are pleased that our responses have addressed these questions.
> >
> > Regarding the remaining concern that our key idea of “pruning by reasoning” is an overclaim, we believe that this assessment should be based on specific evidence and needs further clarification. In our previous visualization results and quantitative analyses, we have demonstrated that the reasoning capabilities of LLMs indeed contribute to token pruning. We would appreciate it if you could provide clear clarifications on the specific aspects where our qualitative and quantitative results are deemed insufficient to support our claims.
> >
> > Furthermore, we would like to emphasize that we are completely open to making minor changes to the title, as allowed by the ICLR guidelines. If there are any alternative suggestions, we would be more than happy to consider them. Our primary goal is to ensure that any differences in wording do not negatively impact the overall evaluation of our paper and the efforts we’ve made to address the concerns during the rebuttal period.
> >
> > We look forward to your feedback and insightful suggestions.

---

### Official Review · Reviewer_27Y2 · 2024-10-31

**Soundness:** 3
**Presentation:** 3
**Contribution:** 2
**Rating:** 5
**Confidence:** 4

**Summary:**

This paper proposes LLM-VTP, a training-free token pruning method for multi-modal video understanding.
Specifically, authors uses the attention score between question token and visual tokens to select semantically related tokens.
Authors integrate LLM-VTP with two Video LLMs, i.e., PLLaVA and Tarsier and conduct experiments on several benchmarks including MVBench, MSVD-QA, MSRVTT-QA, ActivityNet-QA, TGIF-QA and VideoChatGPT-Bench.
Results show that LLM-VTP reduces sequence length of the transformer while maintaining a comparable performance.

**Strengths:**

- The motivation is clear, i.e., using question related visual tokens to prune redundant tokens.
- Experiments are conducted on various benchmarks, demonstrating the effectiveness of LLM-VTP.
- The research topic, i.e., video LLM pruning, has real-world application potential which is valuable.
- On MVBench, the pruned video LLM model even show a better performance compared with the original model.

**Weaknesses:**

-The proposed method uses an encoder to extract all video tokens, a process that is significantly more computationally expensive than reducing tokens in subsequent layers. Reducing tokens post-attention is largely ineffective, as the necessary feature information is already captured by other tokens.
- Is the ablation study of different alpha parameters conducted using PLLaVA? Does similar results appear for Tarsier?
- Does LLM-VTP work on other models like ST-LLM, VideoChat, etc. One concern is that for different models, authors select different hyperparameters for token selection ratio and model layer. This might restrict the generalization ability of LLM-VTP.
- Why Table 3 compares LLM-VTP on Tarsier with other methods on PLLaVA? Is it fair as the alpha and M are different in the two video LLMs.
- For the Figure 4 and Figure 1 (b), the highlighted activation area in different layers are quite similar, then what leads to the performance variance between pruning different layers?

**Questions:**

see the weakness

---

> ### Author Response · Authors · 2024-11-22
> **Response to Reviewer 27Y2 (1/2)**
>
> Thank you for taking the time to review our paper and for providing meticulous and constructive feedback. We are pleased that you found the paper to clearly present the idea and recognize the method’s potential for real-world application.
>
> Please refer to the detailed response below regarding the raised several points.
>
> >**Q1:** The proposed method uses an encoder to extract all video tokens, a process that is significantly more computationally expensive than reducing tokens in subsequent layers. Reducing tokens post-attention is largely ineffective, as the necessary feature information is already captured by other tokens.
>
> **R:** In fact, the parameter size and computational costs of the visual encoder are much smaller than those of the LLM. The table below illustrates this comparison using the PLLaVA and Tarsier models, clearly showing that the LLM incurs a significantly higher computational cost. Thus, our focus on reducing the computational load of the LLM, as outlined in this paper, is both reasonable and justified.
>
> **Table 1: Complexity comparison between vision encoder and LLM layers.**
> |  Method | Vision Encoder Parameter (B) | LLM Parameter (B) | Parameter Ratio (LLM / Vision Encoder) | Vision Encoder FLOPs (T) | LLM FLOPs (T) | FLOPs Ratio (LLM / Vision Encoder) |
> |:--------:|:------------------------------:|:------------------:|:----------------------------------------:|:-------------------------:|:--------------:|:-----------------------------------:|
> | PLLaVA | 0.3                          | 7.1              | 23.3                                   | 6.1                     | 26.0         | 4.3                               |
> | Tarsier| 0.3                          | 7.1              | 23.3                                   | 3.1                     | 57.6         | 18.6                              |
>
> >**Q2:** Is the ablation study of different alpha parameters conducted using PLLaVA? Does similar results appear for Tarsier?
>
> **R:** To clearly demonstrate the impact of different $\alpha$ parameters across various models and datasets, we have included additional results in Figures 3 and 4 of the main paper using both the PLLaVA and Tarsier models. This experiment comparison includes multiple datasets: multi-choice benchmarks (MVBench and VideoMME) and open-ended benchmarks (VCGBench and MSVD-QA). The findings show that both models behave similarly: using $\alpha$=0.2 generally outperforms $\alpha$=0.1 in most cases, as $\alpha$=0.2 provides the model with more information for analysis. Moreover, it is noteworthy that when the layer number reaches 5 or higher, the model’s accuracy tends to saturate, and both $\alpha$=0.1 and 0.2 approach or even exceed the baseline performance. This consistent trend indicates that we can apply similar parameter settings across various datasets and models without relying on extensive optimization and exploration. The results and analysis have been included in Sec. 4.5 of the revised paper.

---

> > ### Comment · Reviewer_27Y2 · 2024-11-25
> >
> > Thank you for your response. The 0.3B parameter encoder yields 6.1 FLOPs, which is comparable to LLM FLOPs. However, increasing the size of the vision encoder and incorporating more video frames will significantly raise the computational cost. Additionally, since pruning is applied in the later layers of LLMs, the efficiency of the current method remains uncertain.

---

> ### Author Response · Authors · 2024-11-22
> **Response to Reviewer 27Y2 (2/2)**
>
> >**Q3:** Does LLM-VTP work on other models like ST-LLM, VideoChat, etc. One concern is that for different models, authors select different hyperparameters for token selection ratio and model layer. This might restrict the generalization ability of LLM-VTP.
>
> **R:** Thank you for your suggestion. In Table 2 below, we present performance and computational cost comparisons on MVBench, VideoMME, and Egoschema, highlighting our model's effectiveness for both long and short videos within the Q-Former architecture. **Consistent with PLLaVA, we perform token pruning at Layer 10 with a retained ratio ($\alpha$) of 0.1 (90\% tokens are pruned) and apply the same retained ratio for Look-M.** Since Llava-PrugMerge is not compatible with the Q-Former architecture for token compression, it is excluded from our comparison. The experimental results suggest that our model better preserves the original baseline accuracy compared to Look-M, and in certain scenarios, it even enhances performance compared to baselines. For example, with VideoChat2, we observe performance improvements in nearly all cases. **These experiments further demonstrate that the hyperparameter settings can be effectively transferred across different models and benchmarks.** Besides, we achieve much lower computational cost than Look-M and the baseline, which demonstrates our efficiency. The experiments have been included in Appendix A.1 of the revised paper.
>
> **Table 2: Performance analysis (top1-acc (\%)) using VideoChat2 and ST-LLM.**
> | Method | FLOPs (T) | MVBench | VideoMME (Short) | VideoMME (Medium) | VideoMME (Long) | VideoMME (Avg) | Egoschema (Full-set) | Egoschema (Sub-set) |
> |--------|:---------:|:------------------:|:-------------------:|:-----------------:|:----------------:|:-----------------------:|:----------------------:|:----------------:|
> | VideoChat2   |   1.1       | 51.1    | 43.1             | **36.9**          | 32.8            | 37.6           | 41.0                  | 50.6                 |
> | VideoChat2 w/ Look-M | 1.1  | 50.0    | 41.9             | 35.0              | 32.6            | 36.5           | 40.9                  | 50.7                 |
> | **VideoChat2 w/ Ours** | **0.5**   | **51.3**| **43.4**         | 36.2              | **34.6**        | **38.1**       | **41.8**              | **52.6**             |
> |                       |         |                  |                   |                 |                |                       |                      |
> | ST-LLM           |  5.4    | **54.9**| **51.4**         | **39.7**          | **34.8**        | **42.0**       | **45.6**              | **56.2**             |
> | ST-LLM w/ Look-M  | 5.4    | 54.0    | 50.6             | 38.0              | 33.1            | 40.6           | 44.5                  | 54.4                 |
> | **ST-LLM w/ Ours**     | **2.1**   | **54.9**   | 51.3             | **39.7**              | **34.8**            | 41.9           | **45.6**                  | 56.0                 |
>
> > **Q4:** Why Table 3 compares LLM-VTP on Tarsier with other methods on PLLaVA? Is it fair as the alpha and M are different in the two video LLMs.
>
> **R:**  In Table 3, `Tarsier w/LLM-VPT (Ours)` is a typo, which should be `PLLaVA w/LLM-VPT (Ours)`. Thanks for pointing out this point, which has been corrected in the revision.
>
> > **Q5:** For the Figure 4 and Figure 1 (b), the highlighted activation area in different layers are quite similar, then what leads to the performance variance between pruning different layers?
>
> **R:** The attention visualizations become similar starting from the 5th layer. However, they are not exactly the same, which results in slight differences in performance accuracy, typically less than 1\%. As illustrated in Figures 3 and 4 of the main paper, the model’s accuracy starts to saturate from layer 5, aligning with the observed visualizations. Notably, the results for layer 1 show a significant deviation from the others. To explore this further, we include a visualization for layer 1 in Figure 5 of the revised paper. This reveals that the attention in layer 1 is quite distinct from the other layers, as it does not effectively focus on key areas, leading to declined accuracy.
>
> Thanks again for your valuable comments and looking forward to your reply.

---

> ### Author Response · Authors · 2024-11-25
> **A Kind Reminder for Reading the Response**
>
> Dear Reviewer 27Y2,
>
> Thank you sincerely for dedicating your time to evaluate our paper. We realize that you may have many commitments at the moment. However, since the discussion period will conclude soon, we would like to gently remind you to share your thoughts on our responses. We are eager to engage in further discussion if you have any additional feedback or concerns regarding our work.
>
> Thank you again for your consideration, and we look forward to hearing from you.
>
> Warm regards,
>
> Authors

---

> ### Author Response · Authors · 2024-11-26
> **Response to Reviewer 27Y2 Comment**
>
> Thanks for your feedback. Below, we provide additional analyses and experimental results to address your new concerns.
>
> > Q1: However, increasing the size of the vision encoder and incorporating more video frames will significantly raise the computational cost.
>
> **R:** In current cutting-edge multi-modal large language models, **it is a common practice to keep the vision encoder size fixed and scale up the LLM sizes, rather than increasing the vision encoder size**. As shown in Table 1 below, we list some representative models along with their vision encoder and LLM sizes. We observe that various works consistently adopt the same vision encoder size while experimenting with language models of varying sizes, which are considerably larger than the vision encoder. Therefore, it is more imperative and necessary to focus on reducing the computational cost of LLMs.
>
> Moreover, incorporating more video frames would indeed raise the computational cost. However, since both vision encoders and LLMs utilize transformer architectures, the increase in computational cost affects the LLMs more significantly due to their larger sizes. Therefore, optimizing the LLM component is crucial for managing overall computational efficiency.
>
> **Table 1: Comparison of Vision Encoder and LLM Sizes.**
>
> | Method               | Vision Encoder Size |     LLM Size      |
> |----------------------|:-------------------:|-----------------|
> | LLaVA [1]            |        0.3B         |    7B / 13B       |
> | Pixtral [2]          |        0.4B         |       12B         |
> | Qwen2-VL [3]         |        0.7B         | 1.5B / 7.6B / 72B |
> | LLaVA-Next-Video [4] |        0.3B         |    7B / 34B       |
> | Video-LLaVA [5]      |        0.3B         |       7B          |
> | PLLaVA [6]           |        0.3B         |  7B / 14B / 34B   |
> | VideoChat2 [7]       |        0.3B         |       7B          |
>
> > Q2: Additionally, since pruning is applied in the later layers of LLMs, the efficiency of the current method remains uncertain.
>
> **R:** In Table 2 below, we present efficiency evaluations conducted on the VideoChatGPT Benchmark, which is the text-generation benchmark with the longest output length among all those we use, making it particularly suitable for evaluating efficiency.
>
> **Table 2: Efficiency comparison of LLMs on VideoChatGPT Benchmark.**
>
> | Method                      | Inference Time (s/per sample) | Inference Speed Up | FLOPs (T) |
> |-|:-:|:-:|:-:|
> | PLLaVA                      | 3.1               | -                  | 26.8      |
> | **PLLaVA w/ LLM-VPT (Ours)**    | **2.2**          | **1.4×**           | **8.8**   |
> | Tarsier                     |      3.6          | -                  | 58.5      |
> | **Tarsier w/ LLM-VPT (Ours)**   |     **2.5**      |      **1.4x**     | **18.0**  |
>
> The results demonstrate that integrating LLM-VPT into existing models yields significant efficiency improvements. Specifically, applying LLM-VPT to PLLaVA reduces the inference time from 3.1 s to 2.2 s, achieving a 1.4× speedup and decreasing FLOPs from 26.8 T to 8.8 T. Similarly, for Tarsier, inference time is reduced from 3.6 s to 2.5 s with a 1.4x speedup, and FLOPs decrease from 58.5 T to 18.0 T. These results confirm that our method enhances efficiency.
>
> We sincierly hope that our response above can address the reviewer's concern and looking forwards to further discussion if there is any remaining concern.
>
> **References**
>
> [1] Liu, Haotian, et al. "Visual instruction tuning." Advances in neural information processing systems 36 (2024).
>
> [2] Pixtral: https://mistral.ai/news/pixtral-12b/.
>
> [3] Wang, Peng, et al. "Qwen2-vl: Enhancing vision-language model's perception of the world at any resolution." arXiv preprint arXiv:2409.12191 (2024).
>
> [4] Zhang, Y., et al. "Llava-next: A strong zero-shot video understanding model." (2024).
>
> [5] Lin, Bin, et al. "Video-llava: Learning united visual representation by alignment before projection." arXiv preprint arXiv:2311.10122 (2023).
>
> [6] Xu, Lin, et al. "Pllava: Parameter-free llava extension from images to videos for video dense captioning." arXiv preprint arXiv:2404.16994 (2024).
>
> [7] Li, Kunchang, et al. "Mvbench: A comprehensive multi-modal video understanding benchmark." Proceedings of the IEEE/CVF Conference on Computer Vision and Pattern Recognition. 2024.

---

> > ### Comment · Reviewer_27Y2 · 2024-11-27
> >
> > I appreciate the authors' detailed response. However, the present method is not limited to vision tokens. For the multimodal LLM, why not prune both vision and text tokens? Mixed token pruning might be more effective. Moreover, If video tokens can be pruned from the vision encoder, the current method can still enhance acceleration.  As reviewer iKaN noted, the current method can also be applied to LLMs without vision tokens. I maintain my initial rating.

---

> ### Author Response · Authors · 2024-11-28
> **Response to New Concerns of Reviewer 27Y2**
>
> Thanks for your response. We would like to provide additional results and analyses to address your new concerns.
>
> > Q1: For the multimodal LLM, why not prune both vision and text tokens? Mixed token pruning might be more effective.
>
> **R:** Our focus on pruning vision tokens rather than implementing mixed token pruning is well justified by the token distribution characteristics in video LLMs. In current video benchmarks, vision tokens constitute an overwhelming majority of the input tokens, while text tokens represent a negligible proportion.
>
> To quantitatively demonstrate this distribution, we conduct a comprehensive analysis across multiple benchmarks:
> - MVBench
> - VideoChatGPT-Bench (VCGBench)
> - VideoQA-Bench (including MSVD, MSRVTT, Activity-Net, and TGIF)
>
> Our analysis covers two models we use in our paper: PLLaVA and Tarsier. The results are summarized in Table 1:
>
> **Table 1: Distribution of Vision and Text Tokens Across Different Benchmarks**
> |   Method   | MVBench Vision Ratio | MVBench Text Ratio | VCGBench Vision Ratio | VCGBench Text Ratio | VideoQA-Bench Vision Ratio | VideoQA-Bench Text Ratio |
> |:-:|:-:|:-:|:-:|:-:|:-:|:-:|
> |   PLLaVA   |        98.6%         |        1.4%        |         99.4%         |        0.6%         |           99.5%           |        0.5%        |
> |  Tarsier  |        99.3%         |        0.7%        |         99.7%         |        0.3%         |           99.8%           |        0.2%        |
>
> As evidenced by the data:
> - For PLLaVA, vision tokens comprise 98.6%-99.5% of total tokens
> - For Tarsier, the proportion is even higher, ranging from 99.3% to 99.8%
>
> Given this overwhelming dominance of vision tokens in the input stream, focusing our pruning strategy exclusively on vision tokens is both practical and efficient. The minimal presence of text tokens (consistently below 1.4%) makes their pruning unnecessary and potentially counterproductive.
>
> > Q2: Moreover, If video tokens can be pruned from the vision encoder, the current method can still enhance acceleration.
>
> **R:** We would like to emphasize that our motivation and the starting point of our design are to explore whether the LLM's own reasoning capabilities can be leveraged to prune tokens that are irrelevant to the question, thereby accelerating the inference of video LLMs. This aspect has not been widely explored in previous studies for video LLMs.
>
> As for pruning tokens from the vision encoder; indeed, there are works [1,2,3,4,5] that have investigated this approach. However, we did not incorporate this idea into our paper for the following reasons:
>
> a. **Training-Free Approach:** We aim for a pruning method that is **training-free** and can be used **without retraining**, **enhancing its practical applicability**. Previous works [1,2,3,4] demonstrate that pruning or merging tokens within the vision encoder requires retraining it to maintain accuracy, which introduces additional computational overhead.
>
> b. **Wide Applicability:** We desire a model that can be readily applied to **different architectures without special modifications**. LLaVA-PruMerge [5] shows that it is only compatible with MLLMs based on frozen CLIP models. For models where the vision encoder has been adapted—such as VideoChat2 and ST-LLM, which are based on Q-former—this approach is no longer effective and cannot be efficiently applied across different models.
>
> Therefore, while pruning tokens from both the video encoder and the LLM is a feasible idea, it would require sacrificing the advantages of being training-free and widely applicable, which contradicts the goals we seek to achieve in this paper.
>
> > Q3: As reviewer iKaN noted, the current method can also be applied to LLMs without vision tokens.
>
> **R:** Our research focuses on exploring the pruning of vision tokens using question tokens. Applying this method to pure-text LLMs differs significantly from the content of our study and does not align with our design philosophy. Nevertheless, we appreciate your suggestion and will consider exploring this direction in future work.
>
> ---
>
> We are pleased to be engaged in a discussion addressing issues such as computational overhead, ablation experiments, generalizability across different models, attention map visualization, and efficiency comparisons. We are delighted that your concerns have been resolved in these areas. For the new concerns you mentioned, we have provided additional analyses above. If you find our response satisfactory, we kindly hope you reconsider your rating.
>
> We look forward to your feedback and are willing to address any further concerns you may have.

---

> ### Author Response · Authors · 2024-11-28
> **References**
>
> **References**
>
> [1] Choudhury, Rohan, et al. "Don't Look Twice: Faster Video Transformers with Run-Length Tokenization." NIPS 2024.
>
> [2] Bolya, Daniel, et al. "Token merging: Your vit but faster." ICLR 2023.
>
> [3] Kong, Zhenglun, et al. "Spvit: Enabling faster vision transformers via latency-aware soft token pruning." ECCV 2022.
>
> [4] Tang, Quan, et al. "Dynamic token pruning in plain vision transformers for semantic segmentation." ICCV 2023.
>
> [5] Shang, Yuzhang, et al. "Llava-prumerge: Adaptive token reduction for efficient large multimodal models." Arxiv 2024.

---

### Official Review · Reviewer_sEPg · 2024-11-03

**Soundness:** 3
**Presentation:** 3
**Contribution:** 3
**Rating:** 6
**Confidence:** 4

**Summary:**

This paper introduces LLM-VTP, which leverages the ability of LLMs to identify useful visual signals for pruning visual features. Experiments on two video LLMs and multiple video understanding tasks validate that this method can effectively reduce the number of tokens without compromising model performance.

**Strengths:**

1. This paper is well written.
2. The proposed method is training-free and small computational burden.

**Weaknesses:**

1. What is the performance of this method in long videos or scenes with drastic changes in temporal?
2. The author should verify the validity of the method on more different types of backbone.
3. The comparative experimental Settings of some tables require further clarification. For example, in Table 1, under the Tarsier method, why is 20% Retained Ratio retained? If 10% is retained, what are the relevant experimental results? In Table 2, under PLLaVA and Tarsier, the Retained Ratio is 20%. So what does that look like for the 10 percent? In Table 3, the comparison methods are all based on PLLaVA. Why does the authors express their own method as Tarsier w/LLM-VPT (Ours)? So what do all the comparisons look like under the same method?
4. The author lacks in-depth analysis of many experimental phenomena. For example "However, the choice of which layer's attention weights to use differs across datasets."

**Questions:**

Please see the weaknesses.

---

> ### Author Response · Authors · 2024-11-22
> **Response to Reviewer sEPg (1/3)**
>
> We are grateful for your thoughtful and thorough review. We are delighted that our paper is found to be well-written and that the proposed method is efficient.
>
> Please refer to the detailed response below regarding the raised several points.
>
> >**Q1:** What is the performance of this method in long videos or scenes with drastic changes in temporal?
>
> **R:** In the table below, we conduct experiments on two popular long video datasets: VideoMME [1] and EgoSchema [2], which span across broad scenarios and diverse temporal dynamics. For VideoMME, we follow the protocol that excludes the use of subtitles. Our experiments use two models, PLLaVA and Tarsier, and compare their accuracy against other methods. In our approach, we set the pruned layer $M$ to 10 and the retained ratio $\alpha$ to 0.1  (90\% of tokens pruned) for both models. Similarly, the retained ratio for Look-M is set to 0.1, while PruMerge+ employs its adaptive compression method, with an average retained ratio of 0.36. The results indicate that our method most effectively preserves the original model’s performance on these datasets compared to Look-M and PruMerge+, and in some cases, it even slightly enhances performance, demonstrating its effectiveness for long video datasets. which demonstrates the effectiveness of our method on long videos. We have included the experimental results and analysis in the revised paper in Appendix A.1.
>
> **Table 1: Performance comparison (top1-acc (\%)) on the long-form datasets, i.e., VideoMME and Egochema.**
>
> |             Method          | **VideoMME （Short）**      |       **VideoMME （Medium）**               |           **VideoMME （Long）**          |     **VideoMME （Avg）**                  | **Egoschema (Subset)**    |     **Egoschema (Fullset)**                 |
> |-----------------------|:-----------------:|:--------------------:|:--------------------:|:--------------------:|:-----------------:|:--------------------:|
> | **PLLaVA**                | 52.2              | **44.4**                 | 36.5                 | **44.4**                 | **47.8**              | **42.6**                 |
> | PLLaVA w/PruMerge+    | 51.3              | 42.8                 | 35.1                 | 43.4                 | 44.5              | 40.7                 |
> | PLLaVA w/Look-M       | 51.0              | 43.2                 | 34.4                 | 42.9                 | 45.6              | 41.5                 |
> | **PLLaVA w/Ours**     | **53.0**              | 43.4                 | **36.7**                 | **44.4**                 | 46.4              | 42.2                 |
> |                       |                   |                      |                      |                      |                   |                      |
> | **Tarsier**               | 53.1              | 40.6                 | 33.1                 | 42.1                 | 56.0              | **49.9**                 |
> | Tarsier w/PruMerge+   | 51.4              | 39.8                 | 36.1                 | 42.4                 | 52.8              | 47.3                 |
> | Tarsier w/Look-M      | 51.1              | 37.6                 | 31.1                 | 39.9                 | 55.5              | 46.4                 |
> | **Tarsier w/Ours**    | **53.4**              | **41.4**                 | **34.9**                 | **43.3**                 | **56.2**              | 47.9                 |

---

> ### Author Response · Authors · 2024-11-22
> **Response to Reviewer sEPg (2/3)**
>
> >**Q2:** The author should verify the validity of the method on more different types of backbone.
>
> **R:** Thank you for your suggestion. To further assess our method's effectiveness across different backbones, we select video LLMs based on the Q-Former projector architecture, specifically VideoChat2 [3] and ST-LLM [4]. This architecture is distinct from the PLLaVA and Tarsier models in our main paper, which employs a simple MLP as the vision projector. In Table 2 below, we present performance and computational cost comparisons on MVBench, VideoMME, and Egoschema, highlighting our model's effectiveness for both long and short videos within the Q-Former architecture. Consistent with PLLaVA, we perform token pruning at Layer 10 with a retained ratio ($\alpha$) 0.1 (90\% tokens are pruned) and apply the same retained ratio for Look-M. Since Llava-PrugMerge is incompatible with the Q-Former architecture for token compression, it is excluded from our comparison. The experimental results suggest that our model better preserves the original baseline accuracy compared to Look-M, and in certain scenarios, it even enhances performance compared to baselines. For example, with VideoChat2, we observe performance improvements in nearly all cases. These experiments further demonstrate that the hyperparameter settings can be effectively transferred across different models and benchmarks. Besides, we achieve much lower computational cost than Look-M and the baseline, which demonstrates our efficiency. The experiments and analysis are included in Appendix A.1 of the revised paper.
>
> **Table 2: Performance analysis (top1-acc (\%)) using VideoChat2 and ST-LLM.**
> |         Method      | FLOPs (T) | MVBench | VideoMME (Short) | VideoMME (Medium) | VideoMME (Long) | VideoMME (Avg) | Egoschema (Full-set) | Egoschema (Sub-set) |
> |-----------------------|:---------:|:------------------:|:-------------------:|:-----------------:|:----------------:|:-----------------------:|:----------------------:|:----------------:|
> | VideoChat2   |   1.1       | 51.1    | 43.1             | **36.9**          | 32.8            | 37.6           | 41.0                  | 50.6                 |
> | VideoChat2 w/ Look-M | 1.1  | 50.0    | 41.9             | 35.0              | 32.6            | 36.5           | 40.9                  | 50.7                 |
> | **VideoChat2 w/ Ours** | **0.5**   | **51.3**| **43.4**         | 36.2              | **34.6**        | **38.1**       | **41.8**              | **52.6**             |
> |                       |         |                  |                   |                 |                |                       |                      |
> | ST-LLM           |  5.4    | **54.9**| **51.4**         | **39.7**          | **34.8**        | **42.0**       | **45.6**              | **56.2**             |
> | ST-LLM w/ Look-M  | 5.4    | 54.0    | 50.6             | 38.0              | 33.1            | 40.6           | 44.5                  | 54.4                 |
> | **ST-LLM w/ Ours**     | **2.1**   | **54.9**   | 51.3             | **39.7**              | **34.8**            | 41.9           | **45.6**                  | 56.0                 |
>
> >**Q3:** The comparative experimental Settings of some tables require further clarification. For example, in Table 1, under the Tarsier method, why is 20% Retained Ratio retained? If 10% is retained, what are the relevant experimental results? In Table 2, under PLLaVA and Tarsier, the Retained Ratio is 20%. So what does that look like for the 10 percent? In Table 3, the comparison methods are all based on PLLaVA. Why does the authors express their own method as Tarsier w/LLM-VPT (Ours)? So what do all the comparisons look like under the same method?
>
>
> **R:** Thank you for your valuable feedback. To clarify the impact of different retained ratio parameters, we provide additional results with a more comprehensive analysis in Figures 3 and 4 of the main paper. These results cover multiple datasets (MVBench and VideoMME for multi-choice benchmarks, and VCGBench and MSVD-QA for open-ended benchmarks), and involve models such as PLLaVA and Tarsier.
>
> Our findings show that token pruning at Layer 5 or beyond, using either a 10\% or 20\% retained ratio, achieves performance comparable to the baseline.  In Table 1, we selected a 20\% retained ratio for the Tarsier model because it offers slightly better performance. However, as Figure 4 (a) illustrates, a 10\% retained ratio also yields results very close to the baseline. It reveals that our method could achieve comparable performance with baseline using a relatively low retained ratio (10%~20%), without introducing many variations. It is a good characteristic for practical application.
>
>  In Table 3, `Tarsier w/LLM-VPT (Ours)` is a typo, which should be `PLLaVA w/LLM-VPT (Ours)`. Thanks for pointing out this point, which has been corrected in the revision.

---

> ### Author Response · Authors · 2024-11-22
> **Response to Reviewer sEPg (3/3)**
>
> >**Q4:** The author lacks in-depth analysis of many experimental phenomena. For example "However, the choice of which layer's attention weights to use differs across datasets."
>
> **R:** Thanks for pointing out this issue. We offer a more comprehensive analysis of the experimental phenomena. As shown in Figures 3 and 4 of the main paper, while selecting different layer attentions may cause mild variations, a consistent trend emerges across datasets: setting attention layer $M$ to 5 or higher yields results nearly equivalent to the baseline model. These findings demonstrate that starting from M=5, the model’s accuracy gradually saturates, enabling performance comparable to the baseline across various models and datasets. This consistent trend suggests that similar parameter settings can be applied across different datasets and models, reducing the need for extensive optimization and exploration. The experiments and analysis have been included in the revised paper in Sec. 4.5.
>
> Thanks again for your valuable comments and looking forward to your reply.
>
> **References**
>
> [1] Fu, Chaoyou, et al. "Video-mme: The first-ever comprehensive evaluation benchmark of multi-modal llms in video analysis." arXiv preprint arXiv:2405.21075 (2024).
>
> [2] Mangalam, Karttikeya, Raiymbek Akshulakov, and Jitendra Malik. "Egoschema: A diagnostic benchmark for very long-form video language understanding." Advances in Neural Information Processing Systems 36 (2023): 46212-46244.
>
> [3] Li, Kunchang, et al. "Mvbench: A comprehensive multi-modal video understanding benchmark." Proceedings of the IEEE/CVF Conference on Computer Vision and Pattern Recognition. 2024.
>
> [4] Liu, Ruyang, et al. "St-llm: Large language models are effective temporal learners." European Conference on Computer Vision. Springer, Cham, 2025.

---

> ### Author Response · Authors · 2024-11-25
> **A Kind Reminder for Reading the Response**
>
> Dear Reviewer sEPg,
>
> We greatly appreciate your valuable time and effort in reviewing our paper. We understand that this may be a busy period for you. As the discussion phase draws to a close, we would like to request your feedback on our responses. If you have any additional comments or questions about our paper, we would be more than happy to discuss them with you in detail.
>
> We look forward to your reply.
>
> Best regards,
>
> Authors

---

> > ### Comment · Reviewer_sEPg · 2024-11-27
> >
> > Thanks to the authors' comprehensive experiments and detailed analysis, my concerns have been answered, so I have raised my score to 6.

---

> > > ### Author Response · Authors · 2024-11-27
> > > **Thanks for your feedback**
> > >
> > > Thanks for your recognition and kind words about our work. Your appreciation inspires us a lot.

---

### Official Review · Reviewer_aFxJ · 2024-11-04

**Soundness:** 4
**Presentation:** 4
**Contribution:** 4
**Rating:** 8
**Confidence:** 4

**Summary:**

LLM-VTP is a training-free visual token pruning method that enhances multi-modal video understanding efficiency by leveraging LLMs' reasoning abilities to selectively prune visual features based on questions, reducing 80%-90% of tokens while maintaining competitive performance.

**Strengths:**

The paper introduces LLM-VTP, a simple yet effective, training-free framework that prunes video tokens by leveraging LLMs' ability to locate question-specific visual tokens from shallow layers, enhancing multi-modal video understanding efficiently without compromising performance, and demonstrating superior results across multiple benchmarks.

**Weaknesses:**

Impaired Relative Position Understanding: The proposed method appears to diminish the model's ability to capture relative positional information. This is indicated by performance drops in the Object Spatial (OS) and Motion Direction (MD) metrics on the MV-Bench dataset for both PLLaVA with the proposed approach and Tarsier with LLM-VTP. Could the authors explain why the method lacks proficiency in calculating relative positions? If the method is effective in this aspect, please provide experimental evidence or references that support this capability.

Evaluation on Long-Form Video Understanding: The paper does not address how the proposed model performs on long-form video understanding tasks, which require the model to handle challenging temporal complexities. Evaluating the method on datasets containing very long-form videos, such as a subset of EgoSchema, and comparing it to the original models like Tarsier, would provide valuable insights into whether localization by attention and pruning is sufficient to induce long temporal relations. Such an evaluation would demonstrate the model's applicability and effectiveness in handling extended video sequences, highlighting its strengths or revealing areas for improvement.

**Questions:**

I asked questions in the weakness section.

---

> ### Author Response · Authors · 2024-11-22
> **Response to Reviewer aFxJ (1/2)**
>
> Thanks for your time and efforts in reviewing our paper and the detailed and insightful feedback. These constructive suggestions are of great help to us. We are pleased with the reviewer's recognition of the model design and extensive experiments in our paper.
>
> Please refer to the detailed response below regarding the raised individual points.
>
> >**Q1:** Impaired Relative Position Understanding: The proposed method appears to diminish the model's ability to capture relative positional information. This is indicated by performance drops in the Object Spatial (OS) and Motion Direction (MD) metrics on the MV-Bench dataset for both PLLaVA with the proposed approach and Tarsier with LLM-VTP. Could the authors explain why the method lacks proficiency in calculating relative positions? If the method is effective in this aspect, please provide experimental evidence or references that support this capability.
>
> **R:** Thank you for this insightful question. Indeed, upon closely examining the performance of PLLaVA and Tarsier on the OS and MD metrics (as shown in Table 1 below), we find that their results are not always better than random guessing. This suggests that both models inherently lack a strong understanding of relative positional information.
>
> To further explore this issue, we select a model (ST-LLM [1]) that consistently performs better than random guessing on both OS and MD. We observe that when using a smaller retained ratio ($\alpha = 0.1$), the model's performance on these metrics declines compared to the baseline. We believe this is because the attention response of LLMs predominantly focuses on the target objects, with less attention given to the surrounding context. When the retained ratio ($\alpha$) is low, the selected tokens are mostly centered on the target objects, leading to a lack of contextual understanding and a decreased ability to perceive relative positions.
>
> Therefore, when we slightly increase $\alpha$ (to $\alpha = 0.2$), the model pays more attention to the surrounding context, effectively enhancing its understanding of relative positional changes. The 0.5\% difference in accuracy on the MD metric corresponds to only one misclassified sample out of two hundred in the MD subset. Hence, by slightly increasing $\alpha$ to 0.2 (while still pruning 80% of the tokens), we can maintain almost the same level of performance in such scenarios.
>
> **Table 1: Performance analysis (top1-acc (\%)) on the OS and MD scenarios.**
> | Method                      | OS   | MD   |
> |-----------------------------|------|------|
> | Random Guess                | 33.3 | 25.0 |
> | PLLaVA                      | 36.0 | 23.5 |
> | Tarsier                     | 31.0 | 44.0 |
> |                             |      |      |
> | ST-LLM                      | 38.5 | 42.5 |
> | ST-LLM w/Ours ($\alpha$=0.1)   | 38.0 | 39.0 |
> | ST-LLM w/Ours ($\alpha$=0.2)   | 38.5 | 42.0 |

---

> ### Author Response · Authors · 2024-11-22
> **Response to Reviewer aFxJ (2/2)**
>
> >**Q2:** Evaluation on Long-Form Video Understanding: The paper does not address how the proposed model performs on long-form video understanding tasks, which require the model to handle challenging temporal complexities. Evaluating the method on datasets containing very long-form videos, such as a subset of EgoSchema, and comparing it to the original models like Tarsier, would provide valuable insights into whether localization by attention and pruning is sufficient to induce long temporal relations. Such an evaluation would demonstrate the model's applicability and effectiveness in handling extended video sequences, highlighting its strengths or revealing areas for improvement.
>
> **R:** Thanks for your suggestions. In Table 2 below, we conduct experiments on two popular long video datasets: VideoMME [2] and EgoSchema [3]. For VideoMME, we follow the protocol that excludes the use of subtitles. Our experiments use two models, PLLaVA and Tarsier, and compare their accuracy against other methods. In our approach, we set the pruned layer $M$ to 10 and the retained ratio $\alpha$ to 0.1 (90\% of tokens are pruned) for both models. Similarly, the retained ratio for Look-M is set to 0.1, while PruMerge+ employs its adaptive compression method, leading to an average retained ratio of 0.36. The results indicate that our method most effectively preserves the original model’s performance on these datasets compared to Look-M and PruMerge+, and in some cases, it even slightly enhances performance, demonstrating its effectiveness for long-form video datasets. We have included the experimental results and analysis in the revised paper in Appendix A.1.
>
> **Table 2: Performance comparison (top1-acc (\%)) on the long-form datasets, i.e., VideoMME and Egochema.**
>
> |            Method           | **VideoMME （Short）**      |       **VideoMME （Medium）**               |           **VideoMME （Long）**          |     **VideoMME （Avg）**                  | **Egoschema (Subset)**    |     **Egoschema (Fullset)**                 |
> |-----------------------|:-----------------:|:--------------------:|:--------------------:|:--------------------:|:-----------------:|:--------------------:|
> | **PLLaVA**                | 52.2              | **44.4**                 | 36.5                 | 44.4                 | **47.8**              | **42.6**                 |
> | PLLaVA w/PruMerge+    | 51.3              | 42.8                 | 35.1                 | 43.4                 | 44.5              | 40.7                 |
> | PLLaVA w/Look-M       | 51.0              | 43.2                 | 34.4                 | 42.9                 | 45.6              | 41.5                 |
> | **PLLaVA w/Ours**     | **53.0**              | 43.4                 | **36.7**                 | **44.4**                 | 46.4              | 42.2                 |
> |                       |                   |                      |                      |                      |                   |                      |
> | **Tarsier**               | 53.1              | 40.6                 | 33.1                 | 42.1                 | 56.0              | **49.9**                 |
> | Tarsier w/PruMerge+   | 51.4              | 39.8                 | 36.1                 | 42.4                 | 52.8              | 47.3                 |
> | Tarsier w/Look-M      | 51.1              | 37.6                 | 31.1                 | 39.9                 | 55.5              | 46.4                 |
> | **Tarsier w/Ours**    | **53.4**              | **41.4**                 | **34.9**                 | **43.3**                 | **56.2**              | 47.9                 |
>
>
> Additionally, we also provide experiments on two network architectures (ST-LLM and VideoChat2) to further demonstrate the effectiveness of our paper. Please refer to [our response to Q2 from Reviewer sEPg](https://openreview.net/forum?id=Acdd83rF1s&noteId=twtUhHHZT8) for more details.
>
> Thanks again for your valuable comments and looking forward to your reply.
>
> **References**
>
> [1] Liu, Ruyang, et al. "St-llm: Large language models are effective temporal learners." European Conference on Computer Vision. Springer, Cham, 2025.
>
> [2] Fu, Chaoyou, et al. "Video-mme: The first-ever comprehensive evaluation benchmark of multi-modal llms in video analysis." arXiv preprint arXiv:2405.21075 (2024).
>
> [3] Mangalam, Karttikeya, Raiymbek Akshulakov, and Jitendra Malik. "Egoschema: A diagnostic benchmark for very long-form video language understanding." Advances in Neural Information Processing Systems 36 (2023): 46212-46244.

---

> ### Author Response · Authors · 2024-11-25
> **A Kind Reminder for Reading the Response**
>
> Dear Reviewer aFxJ,
>
> Thanks for your valuable time and efforts in reviewing our paper. We understand that you may be rather busy during this period. However, as the discussion period will end very soon, we would like to kindly request your feedback on our responses. We would be happy to discuss with you in detail if you have additional comments about our paper.
>
> Look forward to your reply.
>
> Best regards,
>
> Authors

---

> ### Author Response · Authors · 2024-11-29
> **A Gentle Reminder for Reviewing our Response**
>
> Dear Reviewer aFxJ,
>
> We hope this message finds you well and wish you a happy Thanksgiving holiday. We understand that you may be quite busy with your work and family, especially with the holiday season upon us. However, as the extended discussion stage is nearing its end on **December 2nd**, we kindly ask if you could review our responses to see whether they address your concerns. If you are satisfied with our replies, we would be grateful if you could reconsider rating, as this is very important to us. Additionally, please let us know if you have any remaining concerns.
>
> Best,
>
> Authors

---

> ### Author Response · Authors · 2024-12-02
> **A Kind Reminder for Reviewing the Rebuttal**
>
> Dear Reviewer aFxJ,
>
> I hope this message reaches you in good spirits. We are aware that you have a busy schedule. Nonetheless, with the conclusion of the extended discussion phase ending within one day, we kindly request you review our responses to ensure they adequately address your concerns. If our responses address your concerns, we would greatly appreciate it if you could reconsider your rating, as it is important to us.  Moreover, we would like to address any additional concerns you may have.
>
> Warm regards,
>
> Authors

---

> > ### Comment · Reviewer_aFxJ · 2024-12-03
> > **Feedback**
> >
> > Dear Authors,
> >
> > Thank you for presenting additional experiments which resolve the concerns I had. Therefore, I increase the score to accept.

---

> > > ### Author Response · Authors · 2024-12-03
> > > **Thanks for your feedback**
> > >
> > > Dear Reviewer aFxJ,
> > >
> > > Thanks for your positive feedback, which means a lot to us.
> > >
> > >
> > > Best,
> > >
> > > Authors

---

### Official Review · Reviewer_G3XM · 2024-11-04

**Soundness:** 2
**Presentation:** 2
**Contribution:** 2
**Rating:** 5
**Confidence:** 4

**Summary:**

The paper proposed a new method to improve the inference efficiency of LLMs for long video understanding. The core idea of the paper is very straightforward. Due to the redundancy of the video data, the LLMs do not need to process every token from the video sequence. By pruning video tokens, the inference speed of the LLMs can be improved. The criteria for pruning used by the paper is the learned attention weights between the query questions and extracted video tokens. In combination with techniques such as KV cache, the paper is able to achieve 1.4x faster inference speed without sacrificing the accuracy. Experimental results are provided using two baseline LLMs (PLLaVA and Tarsier).

**Strengths:**

The idea of the paper is very intuitive and straightforward, and can also generalize to different LLMs which make the paper more applicable in real-world applications.

The paper provided thorough experimental results and ablation study to justify the effectiveness of the proposed method.

The writing of the paper is clear and easy to follow.

**Weaknesses:**

Based on the results from Figure 3, the choice of the target layer M seems to be very sensitive. It also seems to be data dependent as well. This makes it tricky to pick a propper M without extensive experiments, which defeats the whole purpose of a training-free approach.

The paper motivated the idea from the application of the long video understanding. However, the paper didn’t provide any analysis about how well the method works on long videos.

**Questions:**

Please consider addressing my questions in the Weaknesses section.

---

> ### Author Response · Authors · 2024-11-22
> **Reponse to Reviewer G3XM**
>
> Thank you for your time and effort in reviewing our paper. We are delighted that you found our paper to present a straightforward and intuitive idea and that the writing is clear and easy to understand.
>
> Please refer to the detailed response below regarding the raised several points.
>
> >**Q1:** Based on the results from Figure 3, the choice of the target layer M seems to be very sensitive. It also seems to be data dependent as well. This makes it tricky to pick a proper M without extensive experiments, which defeats the whole purpose of a training-free approach.
>
> **R:** Although selecting different values for M can cause mild variations, there is a consistent trend that emerges across different datasets: setting M to 5 or higher yields results nearly equivalent to those of the baseline model. To better illustrate this, we provide additional experimental results in Figure 3 and Figure 4 of the revised paper, covering various datasets (Multi-choice benchmarks: MVBench and VideoMME, Open-ended Benchmarks: VCGBench and MSVD-QA) and models (PLLaVA and Tarsier). These results demonstrate that starting from M=5, the model’s accuracy gradually saturates, allowing us to achieve performance comparable to the baseline across different models and datasets. This consistent trend indicates that we can apply similar parameter settings across various datasets and models without relying on extensive optimization and exploration.
>
> This analysis is included in Sec. 4.5 of the revision.
>
> >**Q2:** The paper motivated the idea from the application of the long video understanding. However, the paper didn’t provide any analysis about how well the method works on long videos.
>
> **R:** Thanks for the suggestion. We follow the suggestion to conduct experiments on two popular long video datasets, *i.e.*, VideoMME [1] and EgoSchema [2], and report the results in the table below. For VideoMME, we follow the protocol that excludes the use of subtitles. Our experiments use two models, PLLaVA and Tarsier, and compare their accuracy against other methods. In our approach, we set the pruned layer $M$ to 10 and the retained ratio $\alpha$ to 0.1 (90% of tokens are pruned) for both models. Similarly, the retained ratio for Look-M is set to 0.1, while PruMerge+ employs its adaptive compression method, leading to an average retained ratio of 0.36. The results indicate that our method most effectively preserves the original model’s performance on these datasets compared to Look-M and PruMerge+, and in some cases, it even slightly enhances performance, demonstrating its effectiveness for long video datasets. We have included the experimental results and analysis in the revised paper in Appendix A.1.
>
> **Table 1: Performance comparison (top1-acc (\%)) on the long-form datasets, i.e, VideoMME and Egochema.**
> |  Method  | **VideoMME (Short)**      |       **VideoMME （Medium）**               |           **VideoMME （Long）**          |     **VideoMME （Avg）**                  | **Egoschema (Subset)**    |     **Egoschema (Fullset)**                 |
> |-----------------------|:-----------------:|:--------------------:|:--------------------:|:--------------------:|:-----------------:|:--------------------:|
> | **PLLaVA**                | 52.2              | **44.4**                 | 36.5                 | 44.4                 | **47.8**              | **42.6**                 |
> | PLLaVA w/PruMerge+    | 51.3              | 42.8                 | 35.1                 | 43.4                 | 44.5              | 40.7                 |
> | PLLaVA w/Look-M       | 51.0              | 43.2                 | 34.4                 | 42.9                 | 45.6              | 41.5                 |
> | **PLLaVA w/Ours**     | **53.0**              | 43.4                 | **36.7**                 | **44.4**                 | 46.4              | 42.2                 |
> |                       |                   |                      |                      |                      |                   |                      |
> | **Tarsier**               | 53.1              | 40.6                 | 33.1                 | 42.1                 | 56.0              | **49.9**                 |
> | Tarsier w/PruMerge+   | 51.4              | 39.8                 | 36.1                 | 42.4                 | 52.8              | 47.3                 |
> | Tarsier w/Look-M      | 51.1              | 37.6                 | 31.1                 | 39.9                 | 55.5              | 46.4                 |
> | **Tarsier w/Ours**    | **53.4**              | **41.4**                 | **34.9**                 | **43.3**                 | **56.2**              | 47.9                 |
>
>
> In addition, we also conduct experiments on two more network architectures (ST-LLM and VideoChat2) to further demonstrate the effectiveness of our paper. Please refer to our response to Q2 from Reviewer sEPg for more details.
>
> Thanks again for your valuable comments and looking forward to your reply.

---

> > ### Author Response · Authors · 2024-11-22
> > **References**
> >
> > **References**
> >
> > [1] Fu, Chaoyou, et al. "Video-mme: The first-ever comprehensive evaluation benchmark of multi-modal llms in video analysis." arXiv preprint arXiv:2405.21075 (2024).
> >
> > [2] Mangalam, Karttikeya, Raiymbek Akshulakov, and Jitendra Malik. "Egoschema: A diagnostic benchmark for very long-form video language understanding." Advances in Neural Information Processing Systems 36 (2023): 46212-46244.

---

> ### Author Response · Authors · 2024-11-25
> **A Kind Reminder for Reading the Response**
>
> Dear Reviewer G3XM,
>
> Thanks again for your great efforts and insightful comments in reviewing this paper! With the discussion period drawing to a close, we expect your feedback and thoughts on our reply. We look forward to hearing from you, and we can further address unclear explanations and remaining concerns if any.
>
> Regards,
>
> Authors

---

> ### Author Response · Authors · 2024-11-29
> **A Kind Reminder for the End of the Discussion Period**
>
> Dear Reviewer G3XM,
>
> We hope this message finds you well and wish you a happy Thanksgiving holiday. We understand that you may be quite busy with your work and family, especially with the holiday season upon us. However, as the extended discussion stage is nearing its end on **December 2nd**, we kindly ask if you could review our responses to see whether they address your concerns. If you are satisfied with our replies, we would be grateful if you could reconsider rating, as this is very important to us. Additionally, please let us know if you have any remaining concerns.
>
> Best,
>
> Authors

---

> ### Author Response · Authors · 2024-12-02
> **A Kind Reminder for Reviewing the Rebuttal**
>
> Dear Reviewer G3XM,
>
> I hope this message reaches you in good spirits. We are aware that you have a busy schedule. Nonetheless, with the conclusion of the extended discussion phase ending within one day, we kindly request you review our responses to ensure they adequately address your concerns. If our responses address your concerns, we would greatly appreciate it if you could reconsider your rating, as it is important to us.  Moreover, we would like to address any additional concerns you may have.
>
> Warm regards,
>
> Authors

---

> ### Author Response · Authors · 2024-12-03
> **A Kind Reminder for the Feedback**
>
> Dear Reviewer G3XM,
>
> Thank you for the time and effort in reviewing our paper. We sincerely request your feedback on our response, as we believe our rebuttal has effectively addressed your concerns.
>
> Initially, Reviewer sEPg and Reviewer 27Y2 shared similar concerns regarding point (1), specifically the sensitivity of parameter M. We are pleased that our responses (Q4 in **Response to Reviewer sEPg (3/3)**) are considered to have adequately addressed these issues. Similarly, Reviewer aFxJ and Reviewer sEPg initially expressed concerns about point (2), which pertains to experiments on long-form video benchmarks. Our responses (Q2 in **Response to Reviewer aFxJ (2/2)** and Q1 in **Response to Reviewer sEPg (1/3)**) are considered to have satisfactorily resolved these concerns as well.
>
> Given the above, we believe that our responses also well address your concerns, while since we haven't heard from you, we would be keen to know your thoughts on our responses. Your timely reply is critical to us and will leave us with the feasibility to address any further concerns, to ensure a fair assessment of our paper. If there are any additional points you would like us to address, please let us know. Thank you very much.
>
> Warm regards,
>
> Authors

---

> ### Author Response · Authors · 2024-12-04
> **A Gentle Reminder for Updating the Review**
>
> Dear Reviewer G3XM,
>
> I hope this message finds you well. While it is currently not possible for reviewers to post new comments on the paper, you can still update the review and rating if you are satisfied with the response by **clicking the “Edit” button next to your official review**.
>
> Thank you for your consideration.
>
> Warm regards,
>
> The Authors

---

### Author Response · Authors · 2024-11-22
**General Response**

Dear reviewers and area chairs,

We sincerely thank all the reviewers and the area chair for their time and efforts in reviewing our paper. We are delighted that the reviewers find the paper to be well-written (Reviewers G3XM, sEPg, and iKaN), the method to be well-designed and intuitive (Reviewers G3XM, aFxJ, sEPg, and 27Y2), and the experiments to be comprehensive (Reviewers G3XM, aFxJ, 27Y2 and iKaN).

We have provided experiments, analyses, and clarifications to respond to each reviewer’s comments, respectively. We hope that our response has addressed all the concerns and we will be more than happy to address further concerns if there are any.

Below, we would like to highlight the major updates made to the paper during the discussion period.

* **Sec. 4.5**: Following the question raised by Reviewer G3XM, sEPg, and 27Y2, we have included additional experiments (Fig. 3 and Fig. 4) and analysis (Sec. 4.5) on studying the effects of the hyper-parameters.

* **Sec. 4.5**: Following the suggestion proposed by Reviewer iKaN, we have included the comparison of computational cost in Tab. 3, and corresponding analysis in Sec. 4.5.

* **App. A.1**: Following the question raised by Reviewer G3XM, aFxJ, and sEPg, we have included experiments on long-form video benchmarks in Tab. 7 and corresponding analysis in App. A.1.

* **App. A.1**: Following the question raised by Reviewer sEPg and 27Y2, we have included experiments with two additional video LLMs in Tab. 8, and corresponding analysis in App. A.1.

* **App. A.1**: Following the question raised by Reviewer iKaN, we have included a comparison of how well the token pruning methods can maintain baseline results in Tab. 9 and the corresponding analysis in App. A.1.

* **App. A.2**: Following the suggestion proposed by Reviewer iKaN, we have included a side-by-side token selection visualization in Fig. 6, and corresponding analysis in App. A.2.

* **App. A.2**: Following the suggestion proposed by Reviewer iKaN, we have included a comparison of attention map visualization in Fig. 7 and Fig. 8, and corresponding analysis in App. A.2.

Thanks,

Paper6361 Authors

---

### Comment · Area_Chair_Y2kf · 2024-11-24

Dear Reviewers,

This is a friendly reminder that the discussion period will end on Nov 26th (Anywhere on Earth). If you have not already, please take a careful look at the other reviews and author responses, and comment on whether your original rating stands. Thank you.

Best, AC

---

### Comment · Area_Chair_Y2kf · 2024-11-28

Dear reviewers,

This is a friendly reminder that the discussion period has been extended until December 2nd. If you haven’t yet, we kindly encourage you to review the authors' rebuttal and messages at your earliest convenience and confirm whether your comments have been adequately addressed.

We greatly appreciate your service to this process.

Best, AC

---

### Meta-Review · Area_Chair_Y2kf · 2024-12-19

**Metareview:**

To reduce the computational overhead from video redundancy, this paper proposes LLM-VTP, which tries to leverage the reasoning abilities of LLMs to identify and prune visual features by focusing attention on video frames relevant to the question.  The initial scores were 5,5,5,5,6.  Strengths include intuitive idea, clear writing, thorough experiments, and promising results.  Weaknesses include sensitivity to the hyperparameter M, lacking long video analysis, impaired relative positional information, more in-depth analysis needed including different types of backbones, overclaiming statement on LLM reasoning, and limited technical contribution.  Several of these concerns, including long video analysis, more in-depth experiments, sensitivity to M, were adequately addressed, leading to two reviewers increasing their scores for a final score of 5,5,5,6,8.  However, remaining concerns included overclaimed contributions and the approach in terms of pruning design choices.  This is a borderline paper, with split reviews, and largely borderline ratings except for one accept (8) review.  The AC carefully considered the paper, reviews, rebuttal, discussion, and author messages, and unfortunately, feels that the weaknesses outweigh the strengths, and that the paper does not yet meet the bar for publication at ICLR.  The AC would like to encourage the authors to improve the paper and resubmit to a future venue.

**Additional Comments On Reviewer Discussion:**

Strengths include intuitive idea, clear writing, thorough experiments, and promising results.  Weaknesses include sensitivity to the hyperparameter M, lacking long video analysis, impaired relative positional information, more in-depth analysis needed including different types of backbones, overclaiming statement on LLM reasoning, and limited technical contribution.  Several of these concerns, including long video analysis, more in-depth experiments, sensitivity to M, were adequately addressed, leading to two reviewers increasing their scores for a final score of 5,5,5,6,8.  However, remaining concerns included overclaimed contributions and the approach in terms of pruning design choices.  This is a borderline paper, with split reviews, and largely borderline ratings except for one accept (8) review.  The AC carefully considered the paper, reviews, rebuttal, discussion, and author messages, and unfortunately, feels that the weaknesses outweigh the strengths, and that the paper does not yet meet the bar for publication at ICLR.  The AC would like to encourage the authors to improve the paper and resubmit to a future venue.

---

### Decision · Program_Chairs · 2025-01-22

Reject